# Stabilizing the Laughlin state of light: Dynamics of hole fractionalization

**Pavel D. Kurilovich⋆, Vladislav D. Kurilovich,
José Lebreuilly and Steven M. Girvin**

Departments of Physics and Applied Physics, Yale University, New Haven, CT 06520, USA
Yale Quantum Institute, Yale University, New Haven, CT 06520, USA

⋆ pavel.kurilovich@yale.edu

## Abstract

Particle loss is the ultimate challenge for preparation of strongly correlated many-body states of photons. An established way to overcome the loss is to employ a stabilization setup that autonomously injects new photons in place of the lost ones. However, as we show, the effectiveness of such a stabilization setup is compromised for fractional quantum Hall states. There, a hole formed by a lost photon can separate into several remote quasiholes none of which can be refilled by injecting a photon locally. By deriving an exact expression for the steady-state density matrix, we demonstrate that isolated quasiholes proliferate in the steady state which damages the quality of the state preparation. The motion of quasiholes leading to their separation is allowed by a repeated process in which a photon is first lost and then quickly refilled in the vicinity of the quasihole. We develop the theory of this dissipative quasihole dynamics and show that it has diffusive character. Our results demonstrate that fractionalization might present an obstacle for both creation and stabilization of strongly-correlated states with photons.



# 1 Introduction

Since the idea of quantum computing was conceived [1], the possibility of simulating complicated quantum many-body systems was one of the main drives behind the development of a quantum computer. Although a universal quantum computer has so far remained beyond the reach, we are approaching a point at which quantum simulators tailored for specific problems will be able to produce results that cannot be obtained on modern day classical computers [2, 3].

Simulations of topological correlated phases present an important milestone as the complexity of such systems strongly limits the range of analytical and numerical tools for their study. Prototypical states that emerge from the interplay of topology and strong interactions are fractional quantum Hall (FQH) states. Besides bearing a fundamental interest as quantum fluids, FQH states might be practically useful for fault-tolerant quantum computing as they host anyonic excitations. Manipulation of these excitations can realize robust operations on quantum information encoded in a topologically degenerate ground state [4].

A particularly promising direction in simulating bosonic FQH phenomena is quantum simulation with light [5–9]. Within this approach, the toolbox of quantum optics can be used to manipulate individual photons and bring them into a desired state. Achieving FQH states of light requires two cornerstone ingredients: an artificial gauge field, which would simulate the effect of magnetic field for neutral photons, and strong interaction between individual particles. These ingredients are achievable in the microwave domain, in the context of circuit quantum electrodynamics (cQED) [10], and in the optical domain. In a cQED setup, an artificial gauge field can be realized for photons hopping on a lattice of microwave resonators or qubits by employing parametric driving [11–14], non-reciprocal circuit elements [15], magnetic materials [16–18], or complicated geometry in linear circuits [19,20]. In optical systems, gauge fields can be achieved by various means in arrays of optical cavities [21,22], silicon ring resonators [23, 24], and in twisted optical resonators [25]. Strong interactions between the photons may come from the inherent non-linearity of Josephson junctions in cQED [10] and from coupling of light to atoms in optical resonators [26–28]. We note in passing that ingredients for creating bosonic FQH states are also available in the domain of cold atoms [29,30].

Combining gauge fields and interactions with appropriately tuned coherent drives and pulses in theory allows for the realization of few particle FQH states [21, 31–35]. Experimentally, a chiral correlated state was observed in a three-qubit ring [36] and a two-photon Laughlin state was realized in a twisted optical resonator [37]. The main challenge in scaling these schemes to a large particle number comes from inevitable loss of photons into the environment. A promising way around this issue is to use engineered dissipation to stabilize the desired many-body state [38–45]. Recently, this approach was used to stabilize a Mott-insulator state of eight photons in a one-dimensional qubit array [46].

In the context of FQH effect a setup for stabilization of bosonic Laughlin state at $\nu = 1/2$ was proposed in Ref. [47] for a two-dimensional lattice of qubits (along with more complicated FQH states). The idea is to apply a coherent two-photon drive that puts one photon into the system and one into an auxiliary lossy mode at each site of the array. Such a drive realizes an irreversible process of photon injection: it adds photons into the system but is practically incapable of taking them away as this would require the presence of photons in (cold) lossy modes. If the frequency of the drive is tuned to the resonance with the lowest Landau level (LLL), the photons are pumped into the system as long as less than a half of the states in the LLL are occupied. At half-filling, it is impossible to add an extra photon to the LLL without paying energy associated with the interaction between particles. Thus photon-adding transitions become detuned from the resonance and the system stays in the $\nu = 1/2$ state. If a photon is lost somewhere in the system, a hole forms in the Laughlin state but as long as the drive is active this hole is quickly refilled. Overall the system is stabilized in the Laughlin state. The first step for experimentally realizing this stabilization setup was made in a recent work [18] where a Harper-Hofstadter lattice of microwave resonators was coupled to a single transmon qubit. Stabilization setup conceptually similar to that of Ref. [47] was proposed for twisted optical cavities [48].

Although the proof-of-principle works on dissipative stabilization of bosonic FQH states [47–49] have demonstrated the viability of the stabilization setup for small particle numbers, they did not investigate in detail an important challenge for scaling up — fractionalization of holes in FQH states into remote anyons. In the Laughlin state at $\nu = 1/2$, a hole created by the photon loss can be separated in space into two remote quasiholes each of which corresponds to the absence of one half of a photon. The transition refilling a single quasihole is off-resonant as it results in a quasi-particle state with a finite interaction energy; the drive is thus incapable of refilling a quasihole. Consequently, spatially separated quasiholes have a detrimental effect on the quality of preparation of the Laughlin state. This effect was mentioned in Ref. [47], however the probability of formation of isolated quasiholes was estimated to be small and the

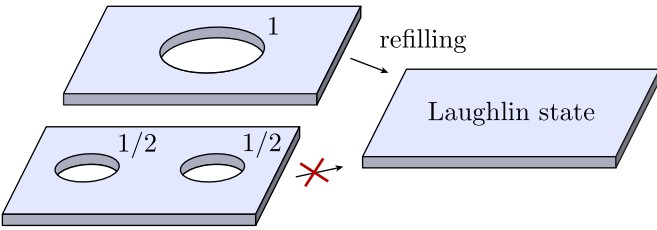

Figure 1: Conceptual picture for the stabilization of the photonic Laughlin state at half-filling. Full hole in the Laughlin state is refilled by adding a photon locally. At the same time two remote quasiholes – which also correspond to the absence of a single photon – cannot be refilled by the stabilization setup. This is because a real (i.e. "bare") photon cannot break into two pieces, in contrast to a hole in the fractional quantum Hall state that can break into two anyons.

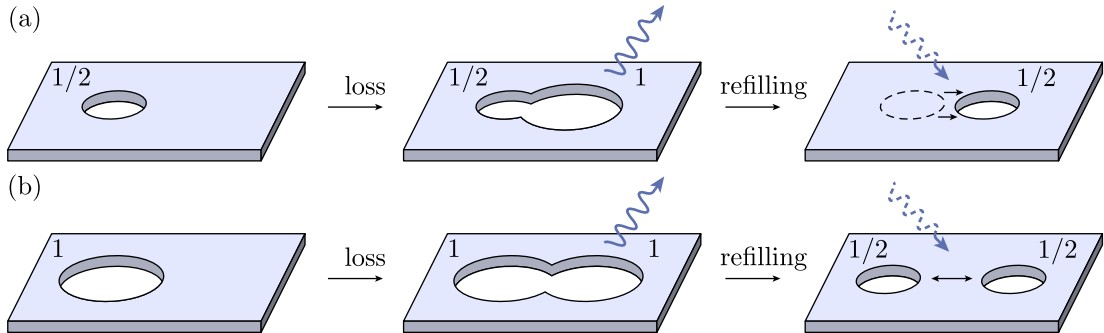

Figure 2: Dissipative dynamics of quasiholes in the stabilized photonic Laughlin state at half-filling. (a) Diffusion of a single quasihole. A photon is first lost in the vicinity of the quasihole and then quickly refiled by the stabilization setup at a different location. As a result the position of the quasihole is shifted in a random direction. (b) A full hole breaks into two stable quasiholes. Again, this requires loss of an additional photon in the vicinity of the initial hole that is followed by a subsequent refilling at a different location.

influence of unpaired quasiholes was neglected.

We demonstrate that unpaired anyons proliferate in the dissipatively stabilized photonic Laughlin state despite the fact that probability of holes to break apart into remote quasiholes is small. In fact, most of the photons missing from the Laughlin state correspond to isolated quasiholes. As a result, the steady state particle number – that should approach half-filling in the ideal case but is generally smaller due to the loss processes – deviates from its target value in a parametrically stronger way than in stabilized correlated states with no fractionalization (such as Mott insulator state [42,46]). We show this by deriving a closed-form expression for the steady state density matrix which allows us to exactly compute the average particle number. We argue that unpaired quasiholes form even if the system is initialized in a pristine Laughlin state and despite the fact that the photons are dissipated and injected locally one by one (as in the model considered in Ref. [47]). This is surprising because only full single-photon holes appear due to loss processes whereas the quasiholes do not posses a Hamiltonian-mediated coherent dynamics (as long as the Landau levels are flat). How then do the holes break apart into the remote quasiholes? We show that dissipation itself makes it possible by endowing the quasiholes with dynamics. Qualitatively, the motion of a quasihole is allowed by a process in which a photon is first lost nearby and then another photon is refilled at a displaced location (see Figure 2). Repeated many times this process leads to a diffusion of a quasihole. We verified this dissipative picture of quasihole mobility analytically and numerically and demonstrated that the diffusion coefficient is proportional to the loss rate.

As a result of hole fractionalization, the preparation of the Laughlin state from vacuum takes a much longer time than the naively expected inverse photon injection rate. Indeed, the stabilization setup pumps photons into the system at random locations. This leads to an abundance of unpaired quasiholes formed during the initialization stage. Abundant quasiholes need to recombine before the steady state is reached. However, this recombination relies on slow diffusive motion of quasiholes mediated by the rare loss processes (loss rate has to be much smaller than the photon injection rate for the stabilization setup to be effective). Therefore, the relaxation of the system is governed by the loss rate as opposed to photon injection rate. As we demonstrate, in a finite size system this behavior is associated with the existence of dark states that cannot be refilled by the stabilization setup. This shows that, counterintuitively, photon loss might be essential for the preparation of the Laughlin state

through engineered dissipation as long as other quasihole motion mechanisms are absent.

Our results demonstrate that fractionalization of holes into quasiholes presents an additional hurdle in preparing the FQH states of light. Although this hurdle does not critically undermine the effort to dissipatively stabilize the FQH states, it raises a question of whether quasihole formation might be suppressed in order to improve the quality of the stabilized state. This would require the development of special protocols that would purge the system of abundant quasiholes. Such protocols might rely either on active manipulation of quasiholes or on the addition of quasihole-trapping potentials. Further research is needed to fully understand how the fractionalization of holes can be suppressed.

The outline of the manuscript is as follows. In Section 2 we describe the model for the stabilization setup. First, we outline the relevant single-particle (see Section 2.1.1) and many-body states (see Section 2.1.2). Then, in Section 2.2 we present the master equation that accounts for loss processes as well as refilling processes induced by the stabilization setup. Starting from Section 3 we present main results of our work. In Section 3 we derive the steady state of the master equation and show that the quality of Laughlin state preparation is diminished due to the fractionalization. In Section 4 we describe two aspects of the dynamics of the system. In Section 4.1 we describe the dissipative dynamics of a single quasihole and in Section 4.2 we consider the dynamics of creating the Laughlin state from the vacuum. Finally, we conclude in Section 5 with an extensive discussion of nuances (such as disorder) that might appear in realistic systems and possible directions for further research.

## 2 Model

Our model for the stabilized photonic Laughlin state consists of two main components. The first is the many-body Hamiltonian that describes the motion of the photons in the artificial magnetic field and their repulsive interaction (see Section 2.1). The second component is the Lindbladian which describes the photon dissipation to the environment as well as the action of the stabilization setup that counteracts this dissipation (see Section 2.2).

### 2.1 Hamiltonian of the system

Many of the proposed realizations of quantum Hall physics with photons require the presence of the lattice (e.g. of qubits or resonators). However, lattice effects greatly complicate the analytical description of the problem [50]. To avoid this complication we assume that the continuum description of the problem is allowed. Such a description is fair for large lattices with small magnetic flux per plaquette or with long-ranged hopping [51].

The purpose of our work is to investigate the effects of hole fractionalization in the bulk of the system. Edge states that are present in a finite planar quantum Hall samples are a natural obstacle towards this goal. Their influence is especially strong on small systems, and only such systems can be treated by our numerics. To eliminate the influence of the edge we consider the problem on a sphere instead of the plane [52]. Rotational invariance present in this approach is an additional useful asset for analytic and numeric investigation. We believe that qualitatively our results remain correct for a large planar sample away from the edge.

All in all, the many-body Hamiltonian that we consider is given by

$$H = \int d^2r \left[ \psi^\dagger(\boldsymbol{r})\hat{T}\psi(\boldsymbol{r}) + g\left(\psi^\dagger(\boldsymbol{r})\psi(\boldsymbol{r})\right)^2 \right]. \tag{1}$$

Here, the integration runs over the surface of the sphere which we assume to have radius $R$. $\psi^\dagger(\boldsymbol{r})$ and $\psi(\boldsymbol{r})$ are bosonic creation and annihilation operators, respectively. They satisfy

the standard bosonic commutation relations $[\psi(\boldsymbol{r}), \psi^\dagger(\boldsymbol{r}')] = \delta(\boldsymbol{r} - \boldsymbol{r}')$, $[\psi(\boldsymbol{r}), \psi(\boldsymbol{r}')] = 0$. Single-particle kinetic energy operator is given by $\hat{T} = m(\boldsymbol{r} \times \boldsymbol{v})^2/2R^2$, with $m$ the effective mass of photons[1]. The velocity operator is given by $\boldsymbol{v} = (-i\hbar\nabla - \boldsymbol{A})/m$ where $\boldsymbol{A}$ is the vector potential describing the uniform artificial magnetic field $\boldsymbol{B} = \nabla \times \boldsymbol{A}$ piercing the surface of the sphere (we choose an unconventional dimensionality of the magnetic field since the photons are chargeless). As follows from the Dirac monopole quantization condition [54], the problem is well defined only if the total flux of the magnetic field equals an integer number $N_\phi$ of flux quanta, $4\pi R^2 n_\phi = N_\phi$, where $n_\phi = B/h$ is the density of flux quanta (here $h = 2\pi\hbar$). Following Ref. [52] we work in a gauge with a cylindrical symmetry,

$$\boldsymbol{A} = \frac{h n_\phi R^2}{r} \cot\theta \, \hat{\varphi} \,. \tag{2}$$

Here $\theta$ and $\varphi$ are polar and azimuthal angles, $\hat{\varphi}$ is the unit vector in the azimuthal direction, and $r$ is the radial distance. Note that vector potential diverges at the poles despite the magnetic field being uniform. These divergences have no physical consequences. Finally, in Eq. (1) constant $g > 0$ is the strength of interaction between the photons. We assume the interaction to be point-like which is analog in the case of a lattice to photons interacting only when at the same site.

In the next two subsections we review the known [52] single-particle and many-body eigenstates of the Hamiltonian (1) that belong to the lowest Landau level (LLL). Such a detailed exposition is prompted by an unusual spherical geometry which we adopt. The discussed eigenstates provide a framework for understanding the analytical results of Secs. 3 and 4.

### 2.1.1 Single-particle states

First, we describe the eigenstates of a single-particle Hamiltonian $\hat{T}$ that belong to the lowest Landau level. Due to the rotational invariance present in the spherical geometry the eigenstates can be characterized by the total angular momentum $S$ and the projection of angular momentum on the $z$-axis, $S_z$. The LLL corresponds to $S = N_\phi/2$ (in units of $\hbar$) and is thus $N_\phi + 1$ times degenerate[2]. All states in the LLL have energy $\hbar\omega_c/2$ with $\omega_c = B/m$ the cyclotron frequency. Explicitly, the wave-function of a state with $S_z = m$ is given by

$$\psi_m(u, v) = \mathcal{C}_m \, u^{N_\phi/2+m} v^{N_\phi/2-m} \,, \tag{3}$$

where $u$ and $v$ parametrize the position on a sphere, $u = \cos(\theta/2)e^{i\frac{\varphi}{2}}$, $v = \sin(\theta/2)e^{-i\frac{\varphi}{2}}$. The normalization factor reads

$$\mathcal{C}_m = \sqrt{\frac{1}{4\pi} \frac{(N_\phi + 1)!}{(N_\phi/2 + m)!(N_\phi/2 - m)!}} \,. \tag{4}$$

Pictorially, the probability density in state $\psi_m$ has a form of a ring-shaped peak centered around $\theta = 2\arctan\sqrt{(N_\phi/2 - m)/(N_\phi/2 + m)}$. As usual, states with different projection of angular momentum are related to each other by action of ladder operators; in terms of variables $u$ and $v$ these operators are given by

$$S_+ = u\partial_v \,, \quad S_- = v\partial_u \,. \tag{5}$$

Operator $S_z$ can also be represented trivially in terms of $u$ and $v$:

$$S_z = (u\partial_u - v\partial_v)/2 \,. \tag{6}$$

---

[1]In context of cQED the effective mass for photons might arise from coherent hopping between the resonators within the 2D lattice. In twisted optical resonators mass appears naturally due to the curvature of the mirrors and cavity length [25, 53].

[2]Notice that the number of states within the LLL on a sphere is larger than its value on a plane – $N_\phi$ – by unity. This discrepancy is known as Wen-Zee shift and its presence results from non-zero curvature of the sphere [55].

### 2.1.2 Many-body states

In this section, we review the many-body states that belong to the LLL and have zero interaction energy. Such states with the different particle numbers are the key ingredients in the dissipative stabilization setup of Ref. [47] on which we focus. We assume that the number $N_\phi$ of flux quanta piercing the sphere is even. Under this condition the Hamiltonian (1) has a unique ground state at half-filling, the Laughlin state (this is not the case for odd $N_\phi$ as we discuss later in Section 4.1).

The non-interacting states within the LLL should simultaneously satisfy three conditions: (i) the wave function has to be symmetric with respect to the permutation of photons due to their bosonic statistics, (ii) the wave function should vanish when particles are at the same position to avoid contact interaction of Eq. (1), (iii) the wave function has to be a polynomial of degree $2S$ in the coordinates $u$ and $v$ of each particle (and not their conjugates). The latter condition guarantees that all particles belong to the LLL in the considered state. Conditions (i)–(iii) can be simultaneously met only below half-filling of the LLL[3], $N \leq N_{1/2}$, where

$$N_{1/2} = N_\phi/2 + 1. \tag{7}$$

For $N > N_{1/2}$ all states either have components in higher Landau levels or non-zero interaction energy. Precisely at half-filling, $N = N_{1/2}$, there is a unique state within the LLL with zero interaction energy, namely the bosonic Laughlin state,

$$\Psi_{\text{LS}} = \prod_{i<j} \left(u_i v_j - u_j v_i\right)^2, \tag{8}$$

(we omit the normalization constants in the description of the many-body states). The indices $i$ and $j$ label different particles, $1 \leq i, j \leq N_{1/2}$.

Let us briefly review the physical properties of the Laughlin state given by Eq. (8). First of all, the uniqueness of the Laughlin state requires it to be rotationally symmetric, $\boldsymbol{S}\Psi_{\text{LS}} = 0$, where $\boldsymbol{S} = \sum_{i=1}^{N_{1/2}} \boldsymbol{S}^i$ is the total angular momentum of all particles. This implies that the density of the photon liquid in the Laughlin state is uniform across the surface of the sphere. Second, the Laughlin state is gapped, i.e. all other states with $N = N_{1/2}$ that belong to the LLL have interaction energy of order of $E_g = g/n_\phi$. Finally, the Laughlin state is also incompressible: it impossible to add one more particle to the Laughlin state without paying energy $\sim \min(E_g, \hbar\omega_c)$ in addition to $\hbar\omega_c/2$ associated with the LLL kinetic energy.

At any particle number below half-filling, $N < N_{1/2}$, there are multiple many-body states with zero interaction energy that belong to the LLL. These states correspond to configurations in which $N_{\text{qh}} = 2(N_{1/2} - N)$ quasiholes are present in the Laughlin state. The factor of two in the latter equation means that for every particle missing from the Laughlin state two quasiholes appear. This is a manifestation of charge fractionalization inherent for the FQH states. For a particular realization of quasihole positions the wave-function is given by

$$\Psi_{\text{qh}}^N = \prod_{i<j} \left(u_i v_j - u_j v_i\right)^2 \times \prod_{l=1}^{N} \prod_{k=1}^{N_{\text{qh}}} \left(u_l \eta_k - v_l \xi_k\right), \tag{9}$$

with $1 \leq i, j \leq N$ and $(u, v) = (\xi_k, \eta_k)$ the positions of the quasiholes. The first multiplier here is similar to that in the Laughlin state; it guarantees that photons do not interact with each other. The second multiplier in Eq. (9) shows that the wave function vanishes whenever one

---

[3]To see this formally, note that conditions (i), (ii) imply that the wave-function has to contain a multiplier $\prod_{i<j} \left(u_i v_j - u_j v_i\right)^2$ (with $i, j$ labeling different particles). However, at $N > N_{1/2}$ such a multiplier by itself would have a higher degree than $2S$ and which violates condition (iii).

of the photons approaches a quasihole. Such dips in the photon density correspond to a net deficit of one half of a photon in each quasihole.

We note that Eq. (9) defines an over-complete basis of quasihole states for a given particle number $N$. To determine the number of linearly independent states we note that any function of the type

$$\Psi_{\text{qh}}^N = \prod_{i<j} \left(u_i v_j - u_j v_i\right)^2 P\left(u_1, v_1, ..., u_N, v_N\right), \tag{10}$$

where $P$ is a symmetric polynomial of the degree $N_{\text{qh}}$, is a valid $N$-particle wave function satisfying conditions (i)–(iii) above. Then, to compute the number of independent quasihole states it is sufficient to count the number of independent polynomials $P$. The result is [56]

$$d_M = \binom{N_{1/2} + N_{\text{qh}}/2}{N_{\text{qh}}}, \tag{11}$$

(notice that $N_{\text{qh}}$ is an even number). Note that $d_0 = d_{2N_{1/2}} = 1$, i.e. the Laughlin state and the state with no photons are non-degenerate. In the limit $N_{1/2} \gg 1$ for fixed $N_{\text{qh}}$, expression (11) reduces to

$$d_{N_{\text{qh}}} \approx \frac{N_{1/2}^{N_{\text{qh}}}}{N_{\text{qh}}!}. \tag{12}$$

This expression has a form of a statistical weight of $N_{\text{qh}}$ bosons placed in $N_{1/2}$ degenerate single-particle states. The fact that the effective number of states is two times smaller than that for full photons in the LLL, $N_{1/2} \approx N_\phi/2$, is due to the artificial charge of quasiholes being one half instead of one.

## 2.2 Stabilization setup

In order to prepare the Laughlin state and preserve it from photon loss, the system is coupled to a stabilization setup based on engineered dissipation (such as the one in Refs. [47,48]). We describe the evolution of the photon density matrix in the presence of such a setup by the following Lindblad master equation:

$$\frac{d\rho}{dt} = -i[H, \rho] + \mathcal{L}_\kappa \rho + \mathcal{L}_\Gamma \rho. \tag{13}$$

Here, $\rho$ is the density matrix of the system and $H$ is its Hamiltonian (1). The superoperator $\mathcal{L}_\kappa$ describes the loss of photons due to the dissipation,

$$\mathcal{L}_\kappa \rho = \kappa \int d^2r \left( \psi(\mathbf{r})\rho\psi^\dagger(\mathbf{r}) - \frac{1}{2}\{\rho, \psi^\dagger(\mathbf{r})\psi(\mathbf{r})\} \right), \tag{14}$$

where $\kappa$ is the photon decay rate. In Eq. (14) we assume that the dissipation of photons happens uniformly across the system in a local way. The superoperator $\mathcal{L}_\Gamma$ describes the action of the stabilization setup that refills the lost photons,

$$\mathcal{L}_\Gamma \rho = \Gamma \int d^2r \left( \tilde{\psi}^\dagger(\mathbf{r})\rho\tilde{\psi}(\mathbf{r}) - \frac{1}{2}\{\rho, \tilde{\psi}(\mathbf{r})\tilde{\psi}^\dagger(\mathbf{r})\} \right). \tag{15}$$

Here, $\Gamma$ is the rate at which the photons are injected into the system (also assumed to be spatially uniform). $\tilde{\psi}(\mathbf{r}) = \mathcal{P}\psi(\mathbf{r})\mathcal{P}$ is the annihilation operator projected on the subspace of the LLL states with zero interaction energy, i.e., the quasihole states of the form (10) with different particle numbers (including $N = 0$ and $N = N_{1/2}$); $\mathcal{P}$ is the corresponding projection operator. We note that $[H, \mathcal{P}] = 0$ because the quasihole states are eigenstates of $H$.

The stabilization setup described by Eq. (15) works in the following way. It injects photons into the LLL for as long as they avoid interacting with one another. Once the Laughlin state is reached, injection of photons ceases. This is highlighted by the presence of projectors in Eq. (15) which do not contain states with more photons than in the Laughlin state. If a full hole is formed in the Laughlin state due to the loss process, photon injection becomes allowed again and the hole is refilled. Thus, for large enough $\Gamma$ the system is stabilized in a state very close to the Laughlin state (we always assume $\Gamma > \kappa$ which is required to make stabilization effective). Physically, selective pumping required for the operation of the stabilization setup might be realized by driving the system incoherently in resonance with the LLL [47]. In this case, processes of photon addition that result in a state with non-zero interaction energy or with components in higher Landau levels are forbidden since they are off-resonant. Importantly, the stabilization setup cannot add photons even for $N < N_{1/2}$ if such a photon-adding process leads to a non-zero interaction energy or finite occupation of higher Ladnau levels[4].

Below we assume that the system starts its evolution in the subspace of the LLL states with zero interaction energy, i.e., that its initial density matrix $\rho_0$ satisfies $\mathcal{P}\rho_0\mathcal{P} = \rho_0$. The latter condition guarantees that the system stays within this subspace at later times, $\mathcal{P}\rho(t)\mathcal{P} = \rho(t)$. Indeed, the discussed subspace is evidently invariant under the action of $\mathcal{L}_\Gamma$ due to the presence of projector operators $\mathcal{P}$ in Eq. (15). It is also invariant under the action of $\mathcal{L}_\kappa$ since the photon loss cannot transfer the system to a state with non-zero interaction energy or with components in higher Landau levels. Therefore, Eqs. (13)–(15) provide a self-contained description of the system within the subspace defined by the projector $\mathcal{P}$. We note that the model is not suited for capturing the behaviour of the system outside of this subspace in physically realistic settings. There, one has to resort to full microscopic models of the stabilization setup [47, 48]. We qualitatively describe some of the effects associated with states that have finite interaction energy (quasiparticle states) in Section 5.1.

We note that Eq. (13) can be obtained from the master equation of Ref. [47] by taking the continuous limit and disregarding all states that have components in higher Landau levels or finite interaction energy. This is justified if the bandwidth of the incoherent driving responsible for the refilling of lost photons is centered around the single-photon LLL energy and is narrow compared to Landau level spacing and interaction energy (in this case the Lorenzian tails of the incoherent photon injection process can be neglected).

Finally, we note that Eq. (13) essentially describes the system coupled to a thermal bath in a Markovian way. The chemical potential $\mu$ and the inverse temperature $\beta$ of this thermal bath satisfy $e^{-\beta(\hbar\omega_c/2-\mu)} = \Gamma/\kappa$ and $e^{-\beta E_g}, e^{-\beta\hbar\omega_c} \ll 1$. The latter condition is required to confine the system to the subspace defined by the projector $\mathcal{P}$.

## 3 Steady state

The stabilization setup is designed to prepare and preserve the Laughlin state (8). However, due to the processes of photon loss the actual steady state $\rho_{\mathrm{st}}$ of master equation (13) – which is defined by the dynamic equilibrium between photon dissipation and injection – deviates from the pure Laughlin state. To find this steady state it is convenient to exchange $\psi(r)$ and $\psi^\dagger(r)$ in equation (14) with $\tilde{\psi}(r)$ and $\tilde{\psi}^\dagger(r)$, respectively. This is justified since the system remains in the subspace defined by projector $\mathcal{P}$ at all times (assuming that it was initialized in

---

[4]To be precise, for a realistic setup of Ref. [47] this statement relies on the presence of the gap above the degenerate quasihole manifold at each particular $N$. To our knowledge, whether such a gap is present in the thermodynamic limit or not is an open question, with recent works claiming that the gap is indeed present [57].

this subspace). Then a direct check shows that

$$\rho_{\text{st}} = \frac{1}{\mathcal{Z}} \mathcal{P} \, (\Gamma/\kappa)^{\hat{N}} \, \mathcal{P} \, , \tag{16}$$

where $\hat{N}$ is the particle number operator and is the exact steady state of master equation (13). Here $\mathcal{Z}$ is a normalization factor that ensures $\text{Tr} \, \rho = 1$.

The steady state $\rho_{\text{st}}$ has a set of notable features. First of all, due to the presence of projectors $\mathcal{P}$ in Eq. (16), the only states that appear in $\rho_{\text{st}}$ are the quasihole states. The probability of a given state depends only on the particle number and for $\Gamma > \kappa$ increases with it. The Laughlin state is thus most probable. The probabilities of individual many-body states with lower particle numbers are smaller. However, these states are degenerate and have a higher statistical weight than the Laughlin state. This leads to a non-trivial interplay between loss and refilling which we elucidate below. Notably, steady state (16) is of the Gibbs form since master equation (13) effectively describes the coupling to a thermal bath.

What metric can be used to quantify the closeness of the steady state (16) to the Laughlin state? An obvious choice of such a metric could be the state infidelity defined as $1 - \mathcal{F}$, where $\mathcal{F} = \langle \Psi_{\text{LS}} | \rho_{\text{st}} | \Psi_{\text{LS}} \rangle$. However, in our case this metric is deficient, as it does not directly translate to the physical properties of the system. Indeed, even the loss of a single photon from the Laughlin state leads to the maximal possible value of infidelity $1 - \mathcal{F} = 1$ because the states now have a different particle numbers. However, far from the point where the photon was lost all observable properties (such as the correlation functions) remain virtually the same as in the Laughlin state.

For our system, a more physically transparent metric for the quality of the state preparation is the relative deviation of the particle number from its target value, $\Delta N / N_{1/2}$, where $\Delta N = N_{1/2} - \langle N \rangle$ ($\langle N \rangle$ is the average number of particles). Indeed, in our model $\Delta N = 0$ unambiguously identifies the Laughlin state[5]. At $0 < \Delta N / N_{1/2} \ll 1$ some photons are missing but the observable properties are still relatively close to that of a Laughlin state.

We now compute $\Delta N / N_{1/2}$ for the steady state (16). Using (16) we find

$$\frac{\Delta N}{N_{1/2}} = \frac{1}{N_{1/2}} \frac{\sum_{n=0}^{N_{1/2}} n \, d_{2n} \, (\kappa/\Gamma)^n}{\sum_{n=0}^{N_{1/2}} d_{2n} \, (\kappa/\Gamma)^n} \, , \tag{17}$$

where the degeneracy factors $d_{2n}$ are given by Eq. (11). For each of $n$ lost photons two quasiholes appear, as indicated by $2n$ in $d_{2n}$. An expression similar to Eq. (17) was previously derived in a recent work [49] where the authors exactly computed the Laughlin state probability in a similar setting. For our purposes, we note that the sums in Eq. (17) can be calculated for a large system with $\Delta N \gg 1$:

$$\frac{\Delta N}{N_{1/2}} \approx \frac{1}{2} \sqrt{\frac{\kappa}{\Gamma}} \, . \tag{18}$$

At the first glance, Eq. (18) seems to contradict simple detailed balance considerations. Indeed, a naive reasoning could run as follows. According to Eq. (13) single photons dissipate from the Laughlin state with rate $\propto \kappa$. A lost photon leaves behind a hole at the corresponding position. If $\Gamma \gg \kappa$, the hole gets quickly refilled by the stabilization setup over the time interval $\propto 1/\Gamma$. Thus $\Delta N / N_{1/2} \sim \kappa/\Gamma$ might be expected [47]. However, this simplistic expectation is not consistent with Eq. (18) which predicts a parametrically larger deviation of the particle number from the Laughlin state. The discrepancy arises because the simplified detailed balance consideration above is oblivious to the effects of the hole fractionalization. In fact, in addition to ephemeral full holes appearing in the Laughlin state there might also exist

---

[5]Because we neglect the presence of high-energy quasiparticle excitations or photons in higher Landau levels.

stable isolated quasiholes. These quasiholes cannot be efficiently refilled by the stabilization setup since it injects photons one-by-one locally [see Eq. (15)] while each quasihole is the absence of only a half of a photon. The fact that the actual relative deviation of the particle number (18) is parametrically larger then the deviation $\sim \kappa/\Gamma$ due to the transient full holes indicates that most of the missing particles in the steady state correspond to separated quasiholes. The abundance of isolated quasiholes in the steady state and the resulting damage to the stabilization setup are the central results of our work.

The proliferation of spatially separated quasiholes in the dissipatively stabilized FQH state of photons should be contrasted with the behaviour of other dissipatively stabilized incompressible phases, which do not exhibit fractionalization of physical properties. A prototypical example of such a phase is the Mott insulator state of photons [46]. There, the effects of hole fractionalization are absent and analysis similar to that leading to Eq. (16) demonstrates $\Delta N/N_{\text{target}} \sim \kappa/\Gamma$ [42]. Here, $N_{\text{target}}$ is the target particle number for the stabilization setup. Formally, the difference between the cases with and without fractionalization stems from different behavior of the degeneracies of subspaces with a given number of particles: fractionalization strongly increases the degeneracy. For example, the state with one lost photon has the degeneracy $\sim N_{\text{target}}$ for Mott insulator and $\sim N_{\text{target}}^2/2!$ for the Laughlin state (the target particle number for the latter is $N_{\text{target}} = N_{1/2}$).

Although the fractionalization increases the deviation of $N$ from its target value, $\Delta N/N_{\text{target}}$ remains a power-law function of $\kappa/\Gamma$. This implies that the dissipative stabilization of the Laughlin state should in principle be experimentally achievable. This conclusion is in line with the previous works [47–49].

From the discussion of the steady state above it might appear that the effects of fractionalization of holes become unimportant for small $\kappa$. Indeed, for $\kappa/\Gamma \to 0$ the deviation from the Laughlin state vanishes. However, this will prove to be incorrect as fractionalization strongly impacts the dynamics described by Eq. (13), especially so when the loss rate is small. Namely, in the latter case fractionalization of holes renders the relaxation dynamics of the system slow.

# 4 Dynamics

Let us now assume that the system is prepared initially in a pure Laughlin state with $\nu = 1/2$. In the course of evolution it should eventually reach steady state (16), in which there is a finite concentration of *isolated* quasiholes $\propto \sqrt{\kappa/\Gamma}$ (where $\kappa$ is the loss rate and $\Gamma$ is the refilling rate). How can this happen given that the photons are dissipated and injected locally one-by-one while each quasihole corresponds to the absence of a half of a photon? To reach $\rho_{\text{st}}$ the full holes should be able to break apart into separate quasiholes, which would then be able to move on their own. This motion cannot be provided by Hamiltonian (1) since the quasiholes are its eigenstates.

The goal of the present section is to show that the motion of quasiholes is induced by the stabilization setup in spite of its local character. Such a dissipative dynamics results from the repeated loss and refilling of a full photon in the vicinity of a quasihole, as was touched upon in the introduction [see Fig. 2]. When $\kappa \ll \Gamma$, the loss process is a bottleneck in this sequence of processes. Therefore, the quasihole dynamics is governed by the loss rate $\kappa$.

In Section 4.1 we show that the motion of a single quasihole bears a diffusive character. To this end we utilize a peculiar parity effect of a $\nu = 1/2$ FQH state on a sphere, where for odd $N_\phi$ there is an unpaired quasihole at half-filling that cannot be refilled by the stabilization setup. This allows us to map the problem of quasihole motion onto the problem of spin diffusion. Then, in Section 4.2 we describe how the stabilization setup prepares the Laughlin state starting from the vacuum. This experimentally relevant problem involves the dynamics

of many quasiholes. Since the quasihole diffusion is controlled by the rate of loss processes, the preparation time of the Laughlin state turns out to be much longer than the naively expected inverse refilling rate.

## 4.1 Diffusion of single quasihole

In order to investigate the motion of quasiholes we consider a sphere pierced by an odd number of magnetic flux quanta. In that case the number of quasiholes can only be odd too. In particular, at half-filling ($N = N_{1/2}^{\text{odd}} = [N_\phi + 1]/2$) there is a single quasihole. The presence of an isolated quasihole renders the state at half-filling degenerate, as the quasihole can have arbitrary spatial location on a sphere. By studying the dynamics within this degenerate single-quasihole manifold it is possible to gain general insights about the motion of quasiholes in our system.

### 4.1.1 Wavefunction of a single-quasihole state

We start the discussion by describing the wave-functions of states with a single quasihole. To formally construct these states for odd $N_\phi$ we can start with $\Psi_{\text{LS}}$ at $N_\phi - 1$ flux quanta (which is an even number) and then increase magnetic flux through the surface of the sphere by one quantum by multiplying this wave function by a quasihole factor,

$$\Psi_{1/2}^{\text{odd}}[u_0, v_0] = \prod_{i=1}^{N_{1/2}^{\text{odd}}} (u_i v_0 - u_0 v_i) \Psi_{\text{LS}}. \tag{19}$$

State (19) corresponds to the presence of a single quasihole at position $(u, v) = (u_0, v_0)$. For any particle $i$ the power of the polynomial in Eq. (19) is higher by one than that in $\Psi_{\text{LS}}$. Thus, state (19) indeed corresponds to $N_\phi$ flux quanta [cf. Eq. (3)]. States of the form (19) with all possible values of $u_0$ and $v_0$ form an over-complete basis in the subspace of single-quasihole states. To construct an orthonormal basis in this subspace we note that the quasihole states possess definite total angular momentum:

$$\mathbf{S}^2 \Psi_{1/2}^{\text{odd}}[u_0, v_0] = \tilde{S}(\tilde{S} + 1) \Psi_{1/2}^{\text{odd}}[u_0, v_0], \tag{20}$$

with $\tilde{S} = N_{1/2}^{\text{odd}}/2 = (N_\phi + 1)/4$. This can be verified by noting that $\Psi_{\text{LS}}$ commutes with $\mathbf{S}^2$ and computing the action of $\mathbf{S}^2$ on the quasihole prefactor $\prod_i (u_i v_0 - u_0 v_i)$ directly. From equation (20) it follows that the orthonormal basis of single-quasihole states consists of states with different projections of angular momentum on a given (e.g., $z$) axis, $m = \tilde{S}, \ldots, -\tilde{S}$. Notably, *many-body* states with one quasihole resemble the *single-particle* LLL states for a particle with half the charge of the opposite sign. Since the sign is opposite, the state with $S_z = \tilde{S}$ corresponds to the presence of a quasihole on the *south* pole of the sphere (in contrast to a single-particle state with the highest $S_z$ which is located close to the *north* pole):

$$|\tilde{S}\rangle \propto \Psi_{1/2}^{\text{odd}}[0, 1] = u_1 \ldots u_{N_{1/2}^{\text{odd}}} \Psi_{\text{LS}}, \quad S_z |\tilde{S}\rangle = \tilde{S} |\tilde{S}\rangle. \tag{21}$$

As usual, the state $|m\rangle$ with $S_z = m$ can be obtained from $|\tilde{S}\rangle$ by acting on it with a many-body version of $S_-$ operator $\tilde{S} - m$ times. Since $S_- \Psi_{\text{LS}} = 0$ it is enough to compute the action of $S_-$ on the prefactor near $\Psi_{\text{LS}}$ in Eq. (21). In this way we find

$$|m\rangle \propto (u_1 \ldots u_{m+\tilde{S}} v_{m+\tilde{S}+1} \ldots v_{N_{1/2}^{\text{odd}}} + \text{permutations}) \Psi_{\text{LS}}. \tag{22}$$

Qualitatively, state $|m\rangle$ describes a ring-shaped dip in photon concentration around

$$\theta = 2 \arctan \sqrt{\frac{\tilde{S} + m}{\tilde{S} - m}},$$

(where $\theta$ is a polar angle on a sphere). Such a dip in concentration totals to a deficit of a half of a photon (as compared to the uniform Laughlin state).

### 4.1.2 Relaxation rates for the quasihole dynamics

The dynamics of the system with odd $N_\phi$ is described by a master equation (13), similarly to the case of even $N_\phi$. The steady state density matrix $\rho_{\mathrm{st}}$ is again given by Eq. (16) (although now the operator $\mathcal{P}$ projects on the subspace of states with an odd number of quasiholes). Thus in the steady state the probabilities of states $|m\rangle$ with a single quasihole are the same for all $m$. This is a manifestation of rotational invariance of the problem.

To single out the dynamics of *one* quasihole, we focus on a limit of a very strong refilling

$$\kappa/\Gamma \ll (N_{1/2}^{\mathrm{odd}})^{-2}, \tag{23}$$

in which the probability of having more than one quasihole in the steady state is small [as can be directly verified by using Eq. (16)]. In this regime, if the system is initialized in a state with a single quasihole, its subsequent evolution boils down to the motion of this quasihole (up to small corrections to the density matrix). This observation, together with the rotational symmetry of the problem, makes it possible to draw analytical conclusions about the evolution of the density matrix.

First, we qualitatively describe the mechanism of motion that allows the quasihole to reach the steady state in which it is distributed uniformly across the sphere. To begin with, we note that the refilling part of the Lindbladian [see Eq. (15)] cannot induce the quasihole dynamics by itself. This is because the photon addition event would lead to a state with a finite interaction energy forbidden within our model. Therefore, a loss process has to happen for the quasihole to move. If the loss happens in the vicinity of the original quasihole position, after the subsequent refilling the quasihole might be displaced [see Fig. 2]. Overall, although the states with a lost photon are ephemeral and the system spends the overwhelming majority of time in a state with a single quasihole, it is the photon loss that governs the motion of the quasihole.

To analyze the motion of a single quasihole we note that under the condition (23), the density matrix can be approximated as

$$\rho(t) = \sum_{m,m'} \rho_{m,m'}(t)|m\rangle\langle m'|. \tag{24}$$

This is due to the fact that the excursions into the manifold with more than one quasihole are short. For the same reason the evolution of the density matrix in Eq. (24) is Markovian. Therefore, $\rho(t)$ can be decomposed in terms of the relaxation eigenmodes $\rho_\lambda$[6]

$$\rho(t) = \rho_{\mathrm{st}} + \sum_\lambda \alpha_\lambda e^{-\lambda t} \rho_\lambda, \quad \lambda > 0, \tag{25}$$

where $\rho_{\mathrm{st}}$ and $\rho_\lambda$ are $(2\tilde{S}+1) \times (2\tilde{S}+1)$ matrices in the subspace of states with a single quasihole; $\alpha_\lambda$ correspond to the decomposition coefficients of the density matrix at $t = 0$ into the relaxation eigenmodes. As we explain below, the rotational invariance allows us to determine the structure of eigenmodes $\rho_\lambda$ *exactly*. A combination of analytical and numerical calculations allows us to analyze the relaxation rates $\lambda$.

We start by deriving the relaxation eigenmodes $\rho_\lambda$. To this end we note that the space formed by operators $|m\rangle\langle m'|$ featured in Eq. (24) can be viewed as a direct product of two

---

[6]Decomposition (25) assumes that the steady state is unique which we verify numerically later.

spins $\tilde{S}$ [58]. Then, from the rotational invariance it follows that the eigenmodes can be classified by the sum of angular momenta of the two spins, $L = 0, 1, \dots, 2\tilde{S}$, and its projection, $M = -L, \dots, L$. Thus, in what follows we label the relaxation modes by these two quantum numbers, $\rho_\lambda = \rho_L^M$. The relaxation rates $\lambda = \Lambda_L$ only depend on $L$ which is another consequence of rotational invariance.

The relaxation eigenmodes $\rho_L^M$ are determined by their commutation relations with the spin-$\tilde{S}$ operators $\tilde{S}_i$ (where $i = \pm, z$):

$$\left[\tilde{S}_z, \rho_L^M\right] = M\rho_L^M \,, \tag{26}$$

$$\left[\tilde{S}_\pm, \rho_L^M\right] = \sqrt{L(L+1) - M(M \pm 1)}\,\rho_L^{M \pm 1} \,. \tag{27}$$

In particular, the steady state is given by $\rho_{\text{st}} \equiv \rho_0^0 = \frac{1}{2\tilde{S}+1}\sum_{m=-\tilde{S}}^{\tilde{S}} |m\rangle\langle m|$; it corresponds to $\Lambda_0 = 0$. To explicitly determine eigenmodes $\rho_L^M$ with higher angular momenta $L > 0$ it is convenient to start with $\rho_L^{-L}$. A direct substitution in Eq. (26) shows that $\rho_L^{-L} \propto \tilde{S}_-^L$ (see Appendix A). Then the remaining modes $\rho_L^M$ with $M > -L$ can be obtained by applying the raising operator $\tilde{S}_+$ via the commutation relation given in Eq. (27). For example, in this way we find modes with $L = 1$:

$$\rho_1^1 = -\tilde{S}_+ \,, \quad \rho_1^0 = \sqrt{2}\tilde{S}_z \,, \quad \rho_1^{-1} = \tilde{S}_- \,. \tag{28}$$

Next, we establish how eigenvalues $\Lambda_L$ with $L \geq 1$ depend on the system size and $L$. We will show below that $\Lambda_L \propto L(L+1)$ as expected for a diffusive process on a sphere. The main idea of the calculation is to relate the relaxation rates $\Lambda_L$ to a certain combination of *classical* transition rates which describe how quickly the system goes from a state $|m\rangle$ to a state $|n\rangle$. This relation, together with the rotational invariance and a series of physically justified assumptions, is sufficient to determine how $\Lambda_L$ depends on $L$.

To introduce the classical transition rates, let us note that the density matrix preserves its diagonal form, $\rho(t) = \sum_m p_m(t)|m\rangle\langle m|$, if it is diagonal initially. This is a consequence of rotational invariance: the coherences between states with different projections of angular momentum do not appear if they are absent initially. Then equation (13) boils down to a classical Markovian rate equation for the probabilities $p_m$ of finding the system in a state $|m\rangle$:

$$\frac{dp_m}{dt} = \sum_{n=-\tilde{S}}^{\tilde{S}} p_n W_{n\to m} - p_m \sum_{n=-\tilde{S}}^{\tilde{S}} W_{m\to n} \,. \tag{29}$$

Here $W_{n\to m}$ is a positive real matrix that determines the transition rate from state $|n\rangle$ to state $|m\rangle$. From the rotational invariance it follows that this matrix is symmetric, $W_{n\to m} = W_{m\to n}$. The matrix elements $W_{n\to m}$ are precisely the aforementioned classical transition rates; we can estimate $W_{n\to m} \propto \kappa$. In general, it is not possible to compute $W_{n\to m}$ analytically. Nonetheless, they will prove useful since *all* low-lying relaxation rates of the system $\Lambda_L$ are determined by a *single* linear combination of $W_{n\to m}$.

We start by deriving an expression for $\Lambda_1$. From the rotational symmetry we know that $\rho_1^0 \propto \tilde{S}_z$. Thus, from Eq. (29) we obtain

$$-\Lambda_1 m = \sum_{n=-\tilde{S}}^{\tilde{S}} n W_{n\to m} - m \sum_{n=-\tilde{S}}^{\tilde{S}} W_{m\to n} \,. \tag{30}$$

Substituting $m = \tilde{S}$ and using the fact that the matrix $W$ is symmetric we find

$$\Lambda_1 = \frac{1}{\tilde{S}}\gamma_1 \,, \quad \gamma_1 = \sum_{k=0}^{2\tilde{S}} k W_{\tilde{S}\to\tilde{S}-k} \,, \tag{31}$$

where the rates $W_{\tilde{S} \to \tilde{S}-k}$ describe the spreading of the quasihole starting from the south pole of the sphere [see Fig. 3(a)]. Equation (31) shows that the rate $\Lambda_1$ is determined by the square of the typical hopping length for the quasihole. Indeed, factor of $k$ in the sum is proportional to $r_{\mathrm{qh}}^2$, where $r_{\mathrm{qh}}$ is the radius of a quasihole wave function with angular momentum $\tilde{S} - k$ (this can be understood by using an analogy with single-particle wave functions of the LLL on a plane). Moreover, $\Lambda_1$ is inversely proportional to the system area $A$ since $\tilde{S} \propto N_\phi \propto A$. Overall, $\Lambda_1 \cdot \tau_{\mathrm{jump}} \propto (\Delta r)^2/A$, where $\Delta r$ is the hopping length, $A$ is the area of the system, and $\tau_{\mathrm{jump}} \sim 1/\kappa$ is the time between subsequent jumps. Therefore, the behavior of $\Lambda_1$ is consistent with a diffusive process.

To further verify the diffusive character of quasihole motion we compute the relaxation eigenvalues with $L > 1$ in a limit of a large system, $\tilde{S} \gg 1$. To this end, we make two physically justified assumptions. First, we assume that for $\tilde{S} \gg 1$ the rates $W_{\tilde{S} \to \tilde{S}-k}$ saturate to a certain thermodynamic limit that corresponds to the case of a stabilized Laughlin state on an infinite plane[7]. Second, we assume that in the thermodynamic limit, the constants $W_{\tilde{S} \to \tilde{S}-k}$ quickly decay with $k$. The decay of $W_{\tilde{S} \to \tilde{S}-k}$ is expected because in order for the quasihole at the south pole to hop a large distance, first a full hole should be formed via a loss process far away from the pole. Then a single photon should be injected into the system, refilling the quasihole at the pole and half of the distant full hole. This process is exponentially suppressed, because the refilling in the stabilization setup is local in space. Under the presented assumptions for $\tilde{S} \gg 1$ we find (see Appendix B for derivation)

$$\Lambda_L \approx \frac{L(L+1)}{2\tilde{S}} \gamma_1 \,, \tag{32}$$

where the corrections are suppressed by an additional factor of $1/\tilde{S}$. Thus, the relaxation eigenvalues $\Lambda_L$ scale with $L$ and $\tilde{S}$ in the same way as the relaxation eigenvalues of the diffusion equation $\partial_t n = D \Delta n$ on a sphere. The structure of the eigenmodes $\rho_L^M$ also parallels that of the spherical harmonics $Y_{LM}$. We conclude that the motion of a single quasihole is indeed diffusive on large spatial scales. The diffusion constant is given by

$$D = \frac{1}{2\pi n_\phi} \gamma_1 \,. \tag{33}$$

Qualitatively, the dynamics of a quasihole represents a sequence of jumps on a length of order of the $1/\sqrt{n_\phi}$. The jump length can be related to the diffusion coefficient as $(\Delta r)^2 = 4D/\gamma_0$ where $\gamma_0 = \sum_k W_{\tilde{S} \to \tilde{S}-k}$ is the total jumping rate.

To back up our assumptions regarding the behavior of $W_{\tilde{S} \to \tilde{S}-k}$ as a function of $k$ and $S$, we find these coefficients numerically in the finite size system (up to $N_{1/2}^{\mathrm{odd}} = 11$). To do this, we solve master equation (13) in the angular momentum representation. This representation is obtained by decomposing the annihilation operators $\psi(\boldsymbol{r})$ in Eqs. (14) and (15) as $\psi(\boldsymbol{r}) = \sum_m \psi_m(\boldsymbol{r}) a_m$, where operator $a_m$ with $m \in \{-N_\phi/2, \ldots, N_\phi/2\}$ destroys a photon in the single particle state $\psi_m(\boldsymbol{r})$ [see Eq. (3)]. This leads to

$$\mathcal{L}_\kappa \rho = \kappa \sum_{m=-S}^{S} \left( a_m \rho a_m^\dagger - \frac{1}{2}\{a_m^\dagger a_m, \rho\} \right), \tag{34}$$

$$\mathcal{L}_\Gamma \rho = \Gamma \sum_{m=-S}^{S} \left( \tilde{a}_m^\dagger \rho \tilde{a}_m - \frac{1}{2}\{\tilde{a}_m \tilde{a}_m^\dagger, \rho\} \right), \tag{35}$$

---

[7] Note that if $\kappa$ remains fixed upon the increase of the system size then $\Gamma$ should also increase to ensure that only one quasihole is present in the system (as is evident from Eq. (23)). If the inequality in Eq. (23) is violated then in a large enough system the diffusion of a single quasihole will be obscured by the presence of other quasiholes.

(a)

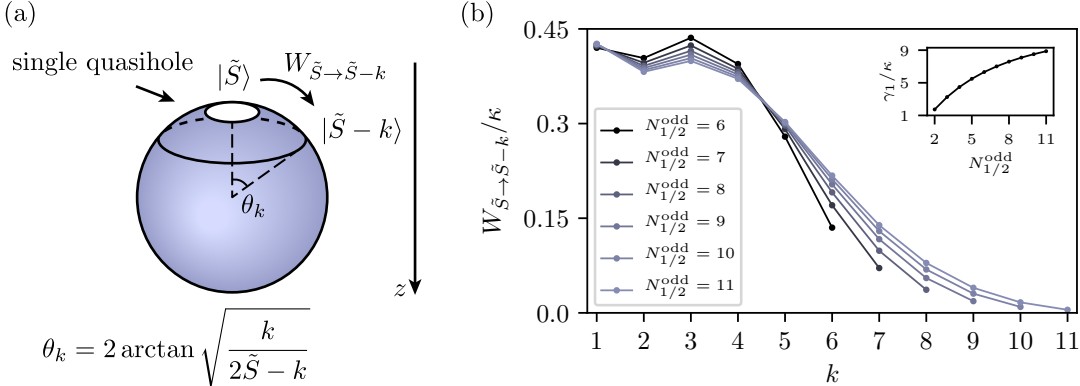

Figure 3: Dynamics of a single quasihole in a stabilized Laughlin state on a sphere ($\nu = 1/2$). The sphere is pierced by an odd number of magnetic flux quanta, $N_\phi$, such that the quasihole is unpaired and cannot be refilled by the stabilization setup. The Hilbert space for the quasihole is formally equivalent to that of a spin $\tilde{S} = (N_\phi + 1)/4$. (a) Due to rotational invariance, the motion of a quasihole is fully characterized by classical hopping rates $W_{\tilde{S} \to \tilde{S}-k}$ from the south pole, state $|\tilde{S}\rangle$, to a state with smaller projection of angular momentum, $|\tilde{S} - k\rangle$. (b) Hopping rate $W_{\tilde{S} \to \tilde{S}-k}$ as a function of $k$ for different number of photons in a Laughlin state $N_{1/2}^{\text{odd}} = (N_\phi + 1)/2$. The plot is obtained with the use of master equation (13) (see details in the main text and in Appendix C). The rate $W_{\tilde{S} \to \tilde{S}-k}$ rapidly decays with the increase of $k$ for $k > 3$. For a fixed $k$, $W_{\tilde{S} \to \tilde{S}-k}$ tends to saturate to a certain limit as system size is increased. The last two statements are consistent with quasihole dynamics being diffusive in a large system. In the inset we show the behavior of $\gamma_1 = \sum_k k W_{\tilde{S} \to \tilde{S}-k}$ with increasing $N_{1/2}^{\text{odd}}$. At large $N_{1/2}^{\text{odd}}$ this quantity should saturate to the diffusion constant for a quasihole (see Eq. (33)). Although for numerically available particle numbers the saturation is incomplete the curve clearly bends down.

where $\tilde{a}_m = \mathcal{P} a_m \mathcal{P}$. This representation allows us to substantially reduce the computational complexity in comparison with using the real-space master equation. Assuming that in the initial state there are no coherences between the states with different $z$-projection of the angular momentum we can disregard them at later times, effectively reducing the size of the Hilbert space. The same is true for coherences between states with different particle numbers $N$. Finally, since we are interested in the motion of a single quasihole, we can truncate the Hilbert space to particle numbers $N = N_{\text{odd}}^{1/2} - 1$ and $N_{\text{odd}}^{1/2}$ assuming that the loss rate is very small, i.e. that condition (23) is well fulfilled. The details of our numeric scheme are outlined in Appendix C.

The result of our numerical procedure is shown in Fig. 3. The figure demonstrates that for each particular $k$ the coefficients $W_{\tilde{S} \to \tilde{S}-k}$ quickly saturate to a thermodynamic limit upon the increase of the system size $N_{1/2}^{\text{odd}}$, as was conjectured previously. Moreover, for all considered particle numbers $W_{\tilde{S} \to \tilde{S}-k}$ decays rapidly[8] with $k$ for $k > 3$. Quantity $\gamma_1$ – directly related to the diffusion coefficient, see Eq. (33) – should also saturate to a thermodynamic limit. Although this saturation is not fully pronounced for particle numbers that we could access, the dependence of $\gamma_1$ on $N_{1/2}^{\text{odd}}$ clearly starts to bend down. We believe that this, combined with the exponential decay of $W_{\tilde{S} \to \tilde{S}-k}$ justifies our claim about the diffusive character of quasihole motion.

---

[8]In fact, for each of the numerically accessible particle numbers $W_{\tilde{S} \to \tilde{S}-k}$ decays with $k$ *quicker* than exponentially. However, we believe this to be a finite-size effect and expect exponential decay in the thermodynamic limit.

## 4.2   Relaxation to the Laughlin state

In this section, we show that the fractionalization of holes renders the relaxation towards the Laughlin state slow. This affects the time it takes to prepare the Laughlin state via the stabilization setup starting from vacuum. To investigate the relaxation, we assume that $N_\phi$ is even so that there are no extra quasiholes at half-filling (in contrast to the previous section). We shall demonstrate that in a small system the relaxation rate scales proportionally to the loss rate, $1/t_{\rm rel} \propto \kappa$. At the same time, one could naively expect that if the system was initialized in the vacuum state it would quickly reach the steady state over the time interval related to the refilling rate, $t_{\rm rel} \propto 1/\Gamma \ll 1/\kappa$. However, such a naive estimate does not hold because upon approaching the Laughlin state the system gets stuck in dark states. These states are characterized by the presence of separated quasiholes that cannot be refilled efficiently by the stabilization setup. To escape a dark state, the system has to wait for a loss process to happen leading to $1/t_{\rm rel} \propto \kappa$. Surprisingly, there are exact dark states in our model: the matrix element for refilling of such states is not just numerically small (e.g., because the quasiholes are spatially separated) but is rather zero identically. Qualitatively, the escape from a dark state corresponds to two remote quasiholes coming together through the motion mechanism discussed in Section 4.1. This makes a process of photon injection that refills the hole possible. At the end of the section, we comment on the relaxation rate in a thermodynamically large system and show that it becomes parametrically slower than that in a small system, $1/t_{\rm rel} \propto \kappa^{3/2}/\Gamma^{1/2}$. These results imply that fractionalization might make the preparation of the Laughlin state through dissipative stabilization challenging.

We start by discussing the origin of dark states in a finite size system. To do that, we investigate the properties of the refilling superoperator $\mathcal{L}_\Gamma$ [see Eq. (35) for the angular momentum representation of the latter[9]]. Considered separately from the loss term of the Lindbladian, this superoperator has the Laughlin state as one of its steady states. In other words, $|\Psi_{\rm LS}\rangle$ vanishes under the action of all jump operators in $\mathcal{L}_\Gamma$, $\tilde{a}_m^\dagger|\Psi_{\rm LS}\rangle = 0$. However, there are other states with a similar property that correspond to a smaller particle number, $N = N_{1/2} - 1$, as we now demonstrate. To see the existence of such dark states it is convenient to classify states with $N = N_{1/2} - 1$ (i.e., the states with two quasiholes) by the $z$-projection of angular momentum. We thus label them as $|M, i_M\rangle$, where $M$ denotes the projection of angular momentum and $i_M = 1, \ldots, n_M$ labels different states with the same value of $M$. The possibility of having $n_M > 1$ is a consequence of fractionalization. Roughly speaking, such a degeneracy might be present because it is often possible to simultaneously increase the angular momentum of one quasihole and lower the angular momentum of the another quasihole preserving the angular momentum. The concrete values of $n_M$ for different $N_{1/2}$ and $M$ can be found by properly counting symmetric polynomials in Eq. (10).

Now we show that among $n_M$ states with a given $M$, $n_M - 1$ states are dark and only one state is bright, i.e., can be refilled to the Laughlin state. First we note that there exists a unique jump operator in $\mathcal{L}_\Gamma$ that can connect $|M, i_M\rangle$ to the Laughlin state: $\tilde{a}_{-M}^\dagger = \mathcal{P}a_{-M}^\dagger\mathcal{P}$. Upon acting with this operator on the Laughlin state, $\tilde{a}_{-M}|\Psi_{\rm LS}\rangle$, we get a linear superposition of states $|M, i_M\rangle$ since it is a two-quasihole state with a correct value of $M$. Then we can rotate the set $|M, i_M\rangle$ in such a way that $|M, 1\rangle$ is proportional to $\tilde{a}_{-M}|\Psi_{\rm LS}\rangle$ and the remaining $n_M - 1$ states are orthogonal to it. The orthogonal states cannot be refilled by any of the jump operators and are thus dark. Therefore, for a given $M$ only one bright state exists. Qualitatively, in this bright state the two qusaiholes have the same angular momentum such that they form a full hole. In contrast, the dark states correspond to remote quasiholes with different angular momenta that sum up to a total of $M$.

The considerations presented above allow us to count the number of bright states $N_{\rm bright}$

---

[9]In contrast to Section 4.1 here $\mathcal{P}$ is the projector on the subspace of quasihole states with even $N_\phi$.

among $d_2 = (N_{1/2} + 1)N_{1/2}/2$ states with $N_{1/2} - 1$ particles. We find

$$N_{\text{bright}} = 2N_{1/2} - 1 = N_\phi \,, \tag{36}$$

consistently with our qualitative interpretation that the bright states are the full holes in the Laughlin state (since the number of the latter is roughly equal to the number of single-particle states in the LLL). These considerations imply that the vast majority of states with $N_{1/2} - 1$ particles are dark and only a small fraction of states $N_{\text{bright}}/d_2 \sim 1/N_{1/2}$ are bright.

We note that there are no evident exact selection rules like that for $N < N_{1/2} - 1$. We believe that the refilling of fractionalized quasihole configurations in this case is actually allowed by $\mathcal{L}_\Gamma$ [as can be seen from numerical solution of Eq. (13)] but is suppressed exponentially in the distance between the quasiholes. However, the detailed investigation of the refilling of states with $N < N_{1/2} - 1$ in a large system exceeds the capabilities of our numeric solution. At the end of this section, we qualitatively study the relaxation process in the limit of a large system. In this case, we expect that the approximate dark states with exponentially suppressed refilling dominate the relaxation.

We mention that the selection rule that is responsible for the existence of the exact dark states with $N = N_{1/2} - 1$ is different from other known selection rules for FQH states, such as the one presented in Ref. [59]. Importantly, the selection rule that we described is not specific for spherical geometry and is thus also applicable to a plane geometry if angular momentum cutoff is assumed. It also applies to a certain lattice models in which Landau levels are flat [51]. It would be interesting to study whether exact selection rules exist in torus geometry [47]. It is also important to mention that the dark states on a sphere (or plane) are robust to the absence of rotational symmetry and they stay intact if the refilling is spatially non-uniform.

Now we focus on the behaviour of system with a moderately small particle numbers in which dark states at $N = N_{1/2} - 1$ have a dominant influence on the relaxation properties. We consider a situation in which both refilling and loss are present and assume that $\kappa/\Gamma \ll (N_{1/2})^{-2}$. In this case the steady state (16) is very close to the Laughlin state, i.e., in the steady state the probability $p_{\text{LS}}$ of finding the system in $\Psi_{\text{LS}}$ is close to unity, $1 - p_{\text{LS}} \ll 1$. We argue that in this case the rate of relaxation to the steady state is $\propto \kappa \ll \Gamma$ instead of $\Gamma$.

To illustrate this, let us consider the system initialized in the vacuum state, $N = 0$, and qualitatively describe its evolution. First, over time $\propto 1/\Gamma$ the system reaches one of the steady states of $\mathcal{L}_\Gamma$. Because the number of dark states with $N_{1/2} - 1$ is much larger than that of the bright states, most likely the system gets stuck in one of the dark states. After that the system has to wait for a time $\propto 1/\kappa$ until it goes back to the manifold with $N = N_{1/2} - 2$ due to the loss process. Such a loss process is followed by a quick refilling process which can potentially bring the system into a bright state with $N_{1/2} - 1$ particles. This allows the system to go to the Laughlin state after an additional successive refilling process. The outlined dynamics of loss and refilling processes is depicted in Figure 4(a). The overall relaxation rate is determined by a bottleneck in the described chain of transitions, which is a slow loss process from $N = N_{1/2} - 1$ to $N = N_{1/2} - 2$. This leads to the relaxation rate $\propto \kappa$.

We note that the described way in which the system escapes from the dark states can be interpreted as a fine-size effect of quasihole diffusion, see Sec. 4.1. For $N_{1/2} \sim 1$ two separated quasiholes present in the dark state can come together in a single diffusion step. This takes time $\propto 1/\kappa$. The resulting bright state contains a full hole that is quickly refilled to the Laughlin state.

To additionally reinforce this qualitative picture of the long relaxation we solve master equation (13) numerically for $N = 7$ and vacuum initial condition. The resulting dependence of $\langle N \rangle$ on time is illustrated in Figure 4(b). The figure demonstrates that after quickly arriving to $N = N_{1/2} - 1$ over time interval $\Delta t \propto 1/\Gamma$ the system gets stuck in a dark state for $\Delta t \propto 1/\kappa$. Only after that it can reach the steady state which is close to the Laughlin state.

To conclude this section, we study the relaxation of the system in the thermodynamic limit, $N_{1/2} \gg \Delta N \gg 1$ (we recall that $\Delta N$ is the average number of photons missing from the Laughlin state). According to the results of Sections 4.1 and 4.2, the dynamics of the system for $\Gamma \gg \kappa$ has the following character. Quasiholes slowly diffuse across the system due to the dissipative dynamics introduced by the stabilization setup. The diffusion coefficient for this motion is proportional to $\kappa$, cf. Eq. (33). The quasiholes are generated from ephemeral full holes breaking apart [see Fig. 2 (b)]. The full holes appear through the loss processes. Whenever two quasiholes come together they might turn into a full hole and thus recombine, i.e., be refilled. The balance between the generation of the quasiholes by the loss processes and the refilling of holes results in a steady state with a relatively large number of quasiholes. In a large system when $\Gamma \gg \kappa$ such dynamics can be phenomenologically captured by a two-component reaction-diffusion model in which one component corresponds to the full holes and another to the isolated quasiholes. The respective system of equation reads

$$\begin{cases} \partial_t n_{\mathrm{h}} = \frac{1}{2}\kappa - \Gamma n_{\mathrm{h}} - \frac{1}{2}c\kappa n_{\mathrm{h}} + c\kappa n_{\mathrm{qh}}^2, \\ \partial_t n_{\mathrm{qh}} = D\nabla^2 n_{\mathrm{qh}} + c\kappa n_{\mathrm{h}} - 2c\kappa n_{\mathrm{qh}}^2. \end{cases} \tag{37}$$

Here, $n_{\mathrm{h}}$ and $n_{\mathrm{qh}}$ are the concentrations of full holes and quasiholes, respectively, measured in units of $n_\phi$; $c$ is a numeric coefficient of order of unity, and $D$ is the diffusion coefficient for the quasiholes [see Eq. (33)]. In Eq. (37) term $\kappa/2$ describes the generation of full holes due to loss processes and $\Gamma n_h$ describes their refilling. The term $\propto \kappa n_h$ describes the ability of full holes to break apart into two quasiholes due to their slow diffusion. The term $\propto \kappa n_{\mathrm{qh}}^2$ corresponds to merging of two quasiholes which creates a full hole. The numeric coefficients in Eq. (37) are chosen to ensure that the concentration of quasiholes is consistent with Eq. (18), see below.

To analyze Eq. (37) we first find the steady state of the system. We obtain the steady-state concentration of quasiholes $n_{\mathrm{qh,st}} = \frac{1}{2}\sqrt{\kappa/\Gamma}$ consistently with Eq. (18). The steady-state concentration of full holes is $n_{\mathrm{h,st}} = \kappa/(2\Gamma) \ll n_{\mathrm{qh,st}}$. Linearizing the system (37) around the steady state and assuming a spatially uniform solution we find the relaxation rate $1/t_{\mathrm{rel}} \propto \kappa^{3/2}/\Gamma^{1/2}$. This shows that in a large system the relaxation rate is even smaller than that $\propto \kappa$ (as found in a small system). The relaxation timescale $t_{\mathrm{rel}}$ corresponds to the time required for a single quasihole to find a partner in course of its diffusion and recombine.

Unfortunately, it is impossible to quantitatively compare the reaction-diffusion model (37) to our numerical results. This is because the system size is comparable to the size of the quasihole for numerically accessible particle numbers while Eq. (37) assumes that fractionalized quasiholes are well-separated. Still, we believe these equations provide a faithful description of the dynamics of a large system.

We note that diffusion-annihilation dynamics of anyons in an open system was also recently studied in one-dimensional Majorana chains [60].

## 5 Discussion and conclusions

To conclude, we investigated the effects of hole fractionalization on the $\nu = 1/2$ Laughlin state of light stabilized against the photon loss. Using the expression for the steady state density matrix which we derived, cf. Eq. (16), we demonstrated that photon number deviates from its target value in a parametrically stronger way than in stabilized many-body states in which the fractionalization is absent. For the Laughlin state the relative deviation of the particle number is $\propto \sqrt{\kappa/\Gamma}$, where $\kappa$ is the photon loss rate and $\Gamma > \kappa$ is the photon refilling rate, cf. Eq. (18). In a dissipatively stabilized Mott insulator of photons — a prototypical correlated bosonic state with no fractionalization — this deviation is only $\propto \kappa/\Gamma$ [42, 46] and is thus much smaller.

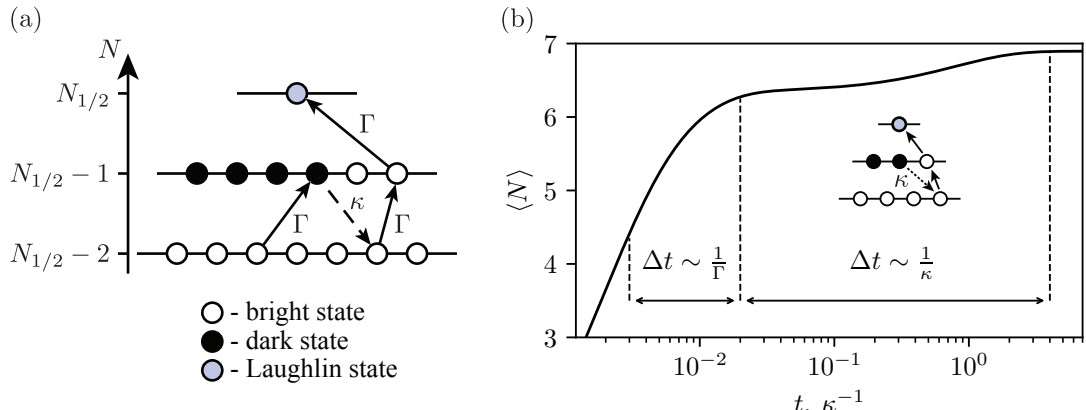

Figure 4: Preparation of the bosonic Laughlin state with a filling factor $\nu = 1/2$. Even number $N_\phi$ of flux quanta pierces the surface of the sphere rendering the Laughlin state non-degenerate. (a) Before reaching the Laughlin state at half filling, $N = N_{1/2}$, the system might get stuck in a dark state with two remote quasiholes (at $N = N_{1/2} - 1$). A loss and a subsequent refilling are needed for the system to escape the dark state and get a chance to be refilled to the Laughlin state. Such a sequence of processes corresponds to two quasiholes coming together due to the dissipative motion mechanism outlined in Fig. 2. (b) Deviation from the Laughlin state, $\Delta N/N$, as a function of $t$ for a small system with $N_{1/2} = 7$ initialized in the vacuum state. After quickly refilling to $N \approx 6$ over time $\propto 1/\Gamma$ (where $\kappa/\Gamma = 4 \cdot 10^{-3}$) the system gets stuck in a configuration with two remote quasiholes for a time $\propto 1/\kappa$. Only after that time quasiholes recombine and the system approaches the Laughlin state. The plot is obtained by solving master equation (13) numerically (see Appendix C for details).

The difference results from the accumulation of separated quasiholes in the stabilized Laughlin state which cannot be effectively refilled by the stabilization setup. The unpaired quasiholes form when full single-photon holes – that appear due to photon loss – break apart. This process is mediated by the dissipative dynamics which is introduced by the stabilization setup, see Fig. 2. We investigated different facets of the dissipative dynamics of quasiholes analytically and numerically. In particular, we showed that the motion of the individual quasiholes is diffusive, with the diffusion coefficient proportional to the photon loss rate $\kappa$, cf. Eq. (33). As a consequence of that, the relaxation rate of the system is much smaller than the refilling rate $\Gamma$ at which photons are injected into the LLL. These results demonstrate that the fractionalization of holes presents an additional challenge for the preparation of fractional quantum Hall states in a bosonic quantum simulator. Below we comment on several important points not discussed in the main text of the manuscript.

## 5.1 Off-resonant injection of photons and optimal value of the refilling rate

First, we discuss a very important nuance behind our model of the stabilization setup which is related to the possibility of the off-resonant photon injection. According to Eq. (18) the average number of lost photons in the steady state decreases with the increase of the photon injection rate $\Gamma$. Thus, it appears that the Laughlin state can be reached with any given precision by making $\Gamma$ sufficiently large. However, this is an oversimplification of our model in which we assume that the stabilization setup can by no means inject photons if the injection requires extra energy either due to the photon repulsion or due to excitation into high Landau levels. In realistic setups there is always a residual rate of adding such high-energy photons. Focusing

on a setup considered in Ref. [47] this rate can be estimated as $\gamma \sim \Gamma \chi^2/\delta^2$ (a similar estimate was given in Ref. [49]), where $\chi$ is the band-width of the drive and $\delta \sim \min(E_g, \hbar\omega_c)$ is the extra energy cost of photon addition (recall that $E_g$ is the typical interaction energy). Thus, if $\Gamma$ is too large – such that $\gamma$ exceeds the loss rate $\kappa$ – a lot of high-energy photons accumulate in the system ruining the effectiveness of the stabilization setup. We conclude that to approach the Laughlin state the refilling rate $\Gamma$ can be neither too small nor too large which implies that there should exist an optimal value $\Gamma_{\text{opt}}$. Here we present an estimate for $\Gamma_{\text{opt}}$ assuming that $\kappa$ and $\delta$ are given. Trivially, the bandwidth of the drive, $\chi$, should be kept as small as possible. In a realistic setting [47] the bandwidth cannot be smaller than $\Gamma$ and thus in what follows we take $\chi \sim \Gamma$ which results in $\gamma \sim \Gamma^3/\delta^2$. Overall, we assume the following hierarchy of the energy scales, $\gamma \ll \kappa \ll \Gamma \sim \chi \ll \delta$.

In a large system, there are two clearly distinct types of defects that make the steady state different from the Laughlin state: fractionalized quasiholes and high-energy photons. The number of quasiholes can be estimated as $N_{\text{qh}} \sim N_{1/2}\sqrt{\kappa/\Gamma}$ [see Eq. (18) and the related discussion]. The number of high-energy photons, $N_{\text{he}}$, is determined by the balance between their injection and photon loss which results in[10] $N_{\text{he}} \sim N_{1/2}\gamma/\kappa$. We expect that the observable properties of the steady state (such as the correlation functions) resemble those of the Laughlin state if, roughly speaking, the total number of defects, $N_{\text{def}} = N_{\text{qh}} + N_{\text{he}}$, is small enough. The minimization of $N_{\text{def}}$ as a function of $\Gamma$ yields the optimal refilling rate

$$\Gamma_{\text{opt}} \sim \kappa^{3/7}\delta^{4/7}. \tag{38}$$

At the optimal refilling rate we obtain

$$N_{\text{def}} \sim N_{\text{qh}} \sim N_{\text{he}} \sim N_{1/2}\left(\frac{\kappa}{\delta}\right)^{2/7}. \tag{39}$$

This can be contrasted with dissipatively stabilized Mott insulator phase of photons [42]. For the latter the optimal concentration of defects scales as $N_{\text{def}}/N_{1/2} \sim (\kappa/\delta)^{1/2}$ and is thus parametrically smaller than that for the stabilized Laughlin state. The increased number of defects in the Laughlin state for $\Gamma = \Gamma_{\text{opt}}$ is a direct consequence of hole fractionalization which is absent in the Mott insulator.

## 5.2 Hamiltonian-induced dynamics of quasiholes

Within our model, the only mechanism of quasihole motion is the loss-mediated dissipative dynamics. Thus, the loss processes are not only a hindrance but also an essential ingredient of the stabilization setup that allows the system to approach the Laughlin state. Without them the system would become stuck indefinitely in states containing remote quasiholes, i.e., the dark states. We note however that in realistic setups, dissipative dynamics is not the only mechanism for quasiholes mobility. Perturbations to the Hamiltonian such as disorder, lattice effects, or long-range interaction between the photons might also lead to the motion of quasiholes thus providing an alternative mechanism by which the system can escape the dark states. We focus on the influence of disorder to demonstrate how the Hamiltonian-mediated dynamics can assist the system in reaching the Laughlin state. We model the disorder potential by a collection of randomly positioned impurities of similar strength,

$$V = \nu_0 \sum_{i=1}^{N_{\text{imp}}} \psi^\dagger(\mathbf{r}_i)\psi(\mathbf{r}_i). \tag{40}$$

---

[10]Note that high-energy photons do not fractionalize into remote quasiparticles. Thus, the square root associated with fractionalization does not appear in the estimate for their number. The quasiparticles do not fractionalize because they are converted into quasiholes with the same rate $\sim \kappa$ as full high-energy photons are destroyed. Therefore, fractionalized quasiparticles do not have additional stability compared to full high-energy photons (as was the case for the quasiholes).

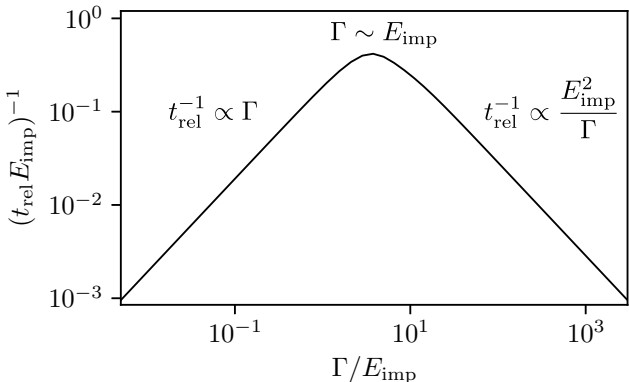

Figure 5: Rate of the relaxation to the Laughlin state, $1/t_{\text{rel}}$, as a function of the refilling rate $\Gamma$ in a small disordered system with no loss, $\kappa = 0$. Due to the latter condition the dissipative dynamics of quasiholes is absent. The plot is produced for $N_{1/2} = 5$ and $N_{\text{imp}} = 100$ assuming that $E_{\text{imp}} \ll \hbar\omega_c$ (where $E_{\text{imp}}$ is the disorder-induced broadening of the LLL and $\omega_c$ is the cyclotron frequency). At $\Gamma \lesssim E_{\text{imp}}$ the relaxation rate is proportional to the refilling rate. In this regime, the quasiholes quickly move due to the disorder and can efficiently recombine; the relaxation rate behaves as if the fractionalization was absent, $t_{\text{rel}}^{-1} \propto \Gamma$. This trend breaks at $\Gamma \sim E_{\text{imp}}$. At higher refilling rates, $\Gamma \gtrsim E_{\text{imp}}$, the dynamics becomes Zeno-blocked and the relaxation rate decreases with increasing refilling rate. Note that if $\kappa = E_{\text{imp}} = 0$ the system would be stuck in dark states indefinitely never reaching the Laughlin state, $1/t_{\text{rel}} = 0$.

Here, $v_0$ is a strength of an individual impurity and $N_{\text{imp}}$ is the total number of impurities. We assume that the positions of impurities are distributed randomly in Poissonian way across the surface of the sphere. If $v_0$ and the concentration $n_{\text{imp}}$ of impurities are small enough, $V$ can be projected on the subspace of quasihole states via the projector $\mathcal{P}$ [see the discussion after Eq. (15) for the definition of $\mathcal{P}$].

As an illustrative idealization, we first assume that the photon loss is absent completely, $\kappa = 0$. We then solve master equation (13) numerically for $N_{1/2} = 5$ in the presence of disorder. The numeric calculation shows that the system reaches the Laughlin state without getting stuck in the dark states even though the loss-mediated dissipative dynamics is turned off. This confirms that in a small system the disorder potential provides a mechanism by which two remote quasiholes can come together to be subsequently refilled. To investigate the disorder-assisted relaxation further, we study the dependence of the relaxation rate on the ratio between the refilling rate $\Gamma$ and the strength of the disorder potential [see Figure 5]. We characterize the latter by energy scale $E_{\text{imp}} = v_0\sqrt{n_{\text{imp}}n_\phi}$ proportional to the disorder-induced broadening of the LLL. When $E_{\text{imp}} \gg \Gamma$ the relaxation rate is $\propto \Gamma$. This linear scaling can be explained qualitatively in the following way. For $E_{\text{imp}} \gg \Gamma$ quasiholes quickly move in the disorder potential. In a small system, pairs of quasiholes come together at a high a rate $\propto E_{\text{imp}}$. Upon every such encounter they have a small chance $\propto \Gamma/E_{\text{imp}}$ to be refilled. Combining the estimates for encounter rate and for the refilling probability we conclude that the relaxation rate indeed scales linearly with $\Gamma$. Interestingly, the linear trend breaks down at $\Gamma \sim E_{\text{imp}}$, and for $\Gamma \gtrsim E_{\text{imp}}$ the relaxation rate *decreases* with $\Gamma$ [see Fig. 5]. In this regime, the decoherence induced by the stabilization setup suppresses the coherent dynamics of quasiholes through Zeno blocking. The relaxation rate corresponding to slow incoherent dynamics of quasiholes in a small system can be estimated as[11] $\sim E_{\text{imp}}^2/\Gamma$, consistent with Fig. 5. We note that for

---

[11]In the regime $\Gamma \gg E_{\text{imp}}$ the relaxation rate is determined by the inverse time it takes for the system to escape

finite loss rate $\kappa$ the relaxation rate ultimately saturates at a value $\sim \kappa$ upon the increase of $\Gamma$ which corresponds to the dissipative dynamics considered in our work.

In a large system, $N_{1/2} \gg 1$, the effect of disorder on the relaxation may differ significantly from the considered case $N_{1/2} \sim 1$ due to Anderson localization. Although, a complete localization is destroyed by the decoherence associated with the stabilization setup, localization physics can still strongly affect the relaxation of the system. A systematic study of the dynamics in the simultaneous presence of localization and engineered dissipation is beyond the scope of the present work.

## 5.3 Stabilization of states with stronger fractionalization

The ability of individual full holes to break apart into several quasiholes becomes even more detrimental for dissipative stabilization of states showing a higher degree of fractionalization. To demonstrate this, we consider the driven-dissipative stabilization of the Laughlin state with the filling fraction $\nu = 1/2n$. The main conceptual difference of this setup with respect to the case $\nu = 1/2$ is that for $\nu = 1/2n$ a full hole can fractionalize into $2n$ quasiholes. Such a system can be described with a master equation similar to (13) although with a different projector $\mathcal{P}$. Now, the latter should project onto the subspace of quasihole states in which the relative angular momentum of each two photons is greater or equal to $2n$. Similarly to how it was done in Section 3, we find the steady-state deviation of the photon number $\Delta N$ from its value in the Laughlin state, $N_\nu$,

$$\frac{\Delta N}{N_\nu} \sim \left(\frac{\kappa}{\Gamma}\right)^{\frac{1}{2n}} . \tag{41}$$

This expression indicates that for a fixed ratio $\kappa/\Gamma$ the difference between the steady-state and the desired Laughlin state becomes parametrically larger upon the increase of $n$. Furthermore, the relaxation to the steady state slows down when $n$ is increased. By doing a reaction-diffusion calculation in the spirit of Section 4.2 we obtain the estimate

$$\frac{1}{t_{\text{rel}}} \sim \kappa \left(\frac{\kappa}{\Gamma}\right)^{1-1/2n} , \tag{42}$$

which shows that the relaxation rate indeed becomes smaller for large $n$. We believe that Eqs. (41) and (42) reflect the general trend that fractionalization makes the stabilization of correlated many-body states inherently challenging. It would be interesting to study how this trend manifests in the dissipative stabilization of non-Abelian FQH states of light, where the quasiholes are associated with additional degrees of freedom. This, however, is beyond the scope of our study.

We note that in practice, the stabilization of Laughlin states with filling fractions $\nu = 1/2n$ would require very precise engineering of interaction between photons. As was mentioned above, to achieve the state with $\nu = 1/2n$, only states in which photons have relative angular momentum larger or equal to $2n$ should be populated by the stabilization setup. This can be achieved by ensuring that first $2n-1$ Haldane pseudo-potentials of the interaction between photons are non-zero and higher pseudo-potentials vanish. Engineering of Haldane pseusdo-potentials in lattice settings (such as FQH state in a lattice of qubits) is discussed in Ref. [61].

---

from a dark state. Physically, such an escape corresponds to two separated quasiholes coming together. For large $\Gamma$, the coherent motion of quasiholes is suppressed and, therefore, two quasiholes can come close only via an incoherent transition process. The rate of this process can be calculated with the help of Fermi Golden rule as $\sim E_{\text{imp}}^2/\Gamma$ (if the system is small enough). Here, $E_{\text{imp}}^2$ is proportional to the squared transition matrix element and factor $1/\Gamma$ originates from decoherence-induced level broadening.

## 5.4 Fractionalization in the stabilized Bose-Hubbard chain

We note that the effects of defect fractionaliztion in driven-dissipative systems are not necessarily unique to topological states such as FQH states. To illustrate that, consider the following extension of the usual Bose-Hubbard model. Suppose there is a lattice of dimension $d$ in which photons can hop between the neighbouring sites. Photons interact with one another via a special type of on-site repulsion that is present only when more than two photons occupy the same site [62, 63]. We assume that such a repulsion is strong enough so that the possibility of having more than two photons per site can be neglected. We also assume that two-photon loss can happen at each site with a rate $\kappa$ [39, 64–66]. Next, there is a stabilization setup that is intended to keep the system in a state in which there are two photons at each site of the lattice. To this end, the scheme attempts to inject pairs of photons at each site with a rate $\Gamma$. Due to the interaction, such an injection is possible only when the site is empty. The presence of a single photon at a given site blocks both the injection and the loss.

In many qualitative aspects, this model resembles the dissipatively stabilized Laughlin state with the filling fraction $\nu = 1/2$. Empty sites can be associated with full holes in the Laughlin state, since the stabilization setup can easily inject photons at them. Sites with a single photon present resemble isolated quasiholes – they too cannot be refilled. To emphasize the similarity, we find the steady-state deviation of the photon number, $\Delta N$, from its target value, $N_{\text{target}}$ (the latter is equal to twice the number of sites). Similarly to the FQH state, we obtain $\Delta N \propto N_{\text{target}} \sqrt{\kappa/\Gamma}$, cf. Eq. (18). The square root behavior indicates that most of the missing particles in the steady state correspond to sites with a single photon. In the same way, most of the particles missing from the Laughlin state are isolated quasiholes.

We emphasize that the fractionalization in a Bose-Hubbard lattice is facilitated by the coherent inter-site hopping, which allows pairs of photons to separate. It would be interesting to analyze the dynamics of the hopping-mediated fractionalization in detail. This, however, is beyond the scope of our work.

We note that in a fermionic case the dynamics of a Hubbard model with a two-body loss was studied in a recent work [67].

## 5.5 Toric geometry

We mention one more avenue for further research. Namely, it would be interesting to study the dissipatively stabilized FQH states on manifolds with topology different from that of a sphere, e.g., on a torus. The toric geometry was considered in Ref. [47] though the effects of hole fractionalization – which are in the focus of our study – were not systematically investigated.

In the context of our work, the main difference between the torus and the sphere is associated with the topological degeneracy of the Laughlin state in the former case. For $\nu = 1/2$ the Laughlin state on a torus is doubly degenerate. The degenerate states are topologically protected, i.e., they cannot be coupled by local perturbations. Thus, for example, if a photon is lost from the Laughlin state and then quickly refilled by the stabilization setup at the same position, the system does not transition between the degenerate states. Such a transition requires a highly non-local process the simplest example of which is the following. First, a pair of quasiholes should form due to the loss of a photon. Then, the two quasiholes have to separate, make a non-contractible loop around the torus, and come back together. Only then, upon a subsequent refilling of the quasiholes, does the system end up in a state orthogonal to the initial one. This suggests that the relaxation rate decreases with increasing system size. However, at finite concentration of quasiholes the above picture might be significantly different. For example, for a two-dimensional toric code coupled to a thermal bath – another model with a topological degeneracy and dynamic, deconfined anyons – is known to be independent of the system size in a thermodynamic limit [68]. It would thus be interesting to determine how the

relaxation rate between topologically degenerate Laughlin states scales with the system size and loss rate within our model, where the motion of quasiholes is mediated by subsequent loss and refilling processes.

## Acknowledgements

We acknowledge insightful discussions with Leonid Glazman and Moshe Goldstein. We also thank Alexey Khudorozhkov for pointing us to Ref. [68].

**Funding information**    JL was supported by Yale University through a Prize Postdoctoral Fellowship in condensed matter theory.

## A   Relaxation eigenmodes for a single quasihole

In this appendix, we find the structure of the relaxation eigenmode $\rho_L^M$ describing the motion of a single quasihole on a sphere. To start with, we show that $\rho_L^{-L} = \tilde{S}_-^L$. This can be done by verifying two relations [58]:

$$[\tilde{S}_-, \rho_L^{-L}] = 0 \,, \qquad [\tilde{S}_z, \rho_L^{-L}] = -L\rho_L^{-L} \,. \tag{43}$$

The first equation is evidently true for $\rho_L^{-L} = S_-^L$. The second equation can be checked directly by using $[\tilde{S}_z, \tilde{S}_-] = -\tilde{S}_-$. Indeed, by applying this commutator $L$ times we find $S_z S_-^L = -L S_-^L S_z$ and thus

$$[\tilde{S}_z, \tilde{S}_-^L] = -L\tilde{S}_-^L \,. \tag{44}$$

The eigenmode with arbitrary $M$ can be obtained by sequentially applying the raising operator:

$$\rho_L^M \propto \underbrace{[\tilde{S}_+, ... [\tilde{S}_+, [\tilde{S}_+, \tilde{S}_-^L]]...]}_{L+M \text{ times}} \,. \tag{45}$$

## B   Relaxation eigenvalues for a single quasihole

In this appendix, we derive Eq. (32) for the relaxation rates $\Lambda_L$ that characterize the motion of a single quasihole on a sphere. This equation — together with the structure of the modes $\rho_L^M$ — demonstrates that the motion of a quasihole is diffusive in a large enough system.

For the sake of deriving Eq. (32), it is useful to first consider the mode with $M = 0$. This mode is purely diagonal in the $\tilde{S}_z$ basis. Indeed, expression (45) for $M = 0$ contains an equal number of lowering and raising operators and thus $\rho_L^0$ conserves the $z$-projection of the angular momentum. The fact that the mode is diagonal allows us to apply the classical master equation (29), leading to the relation between $\Lambda_L$ and the classical hopping rates. The example of this procedure for $\rho_1^0$ was demonstrated in Section 4.1 of the main text. The explicit substitution of $\rho_L^0$ into master equation (29) yields

$$-\Lambda_L \rho_{L,mm}^0 = \sum_{n=-\tilde{S}}^{\tilde{S}} \rho_{L,nn}^0 W_{n\to m} - \rho_{L,mm}^0 \sum_{n=-\tilde{S}}^{\tilde{S}} W_{m\to n} \,, \tag{46}$$

where $\rho^0_{L,mm}$ is the $m$-th diagonal matrix element of $\rho^0_L$ and

$$\rho^0_L \propto \underbrace{[\tilde{S}_+, ...[\tilde{S}_+, [\tilde{S}_+, \tilde{S}^L_-]]...]}_{L \text{ times}}. \tag{47}$$

To proceed, we note that the relaxation mode (47) can be expanded around the south pole of the sphere ($m = \tilde{S}$) as

$$\rho^0_{L,(\tilde{S}-n)(\tilde{S}-n)} \propto 1 - \alpha_L \frac{n}{\tilde{S}} + O\left(\frac{n^2}{\tilde{S}^2}\right), \tag{48}$$

where $\alpha_L$ is related to the logarithmic derivative of $\rho^0_L$,

$$\alpha_L = \tilde{S} \frac{\rho^0_{L,\tilde{S}\tilde{S}} - \rho^0_{L,(\tilde{S}-1)(\tilde{S}-1)}}{\rho^0_{L,\tilde{S}\tilde{S}}}. \tag{49}$$

We note that the matrix element $\rho^0_{L,(\tilde{S}-1)(\tilde{S}-1)}$ here can be rewritten as

$$\rho^0_{L,(\tilde{S}-1)(\tilde{S}-1)} = \langle \tilde{S}-1| \underbrace{[\tilde{S}_+, ...[\tilde{S}_+, [\tilde{S}_+, \tilde{S}^L_-]]...]}_{L \text{ times}} |\tilde{S}-1\rangle$$

$$= \langle \tilde{S}-1|\tilde{S}^L_+\tilde{S}^L_-|\tilde{S}-1\rangle - L\langle \tilde{S}-1|\tilde{S}^{L-1}_+\tilde{S}^L_-S_+|\tilde{S}-1\rangle = \frac{1}{2\tilde{S}}\langle \tilde{S}|\tilde{S}^{L+1}_+\tilde{S}^{L+1}_-|\tilde{S}\rangle - L\langle \tilde{S}|\tilde{S}^L_+\tilde{S}^L_-|\tilde{S}\rangle$$

$$= \left(\frac{1}{2\tilde{S}}\langle \tilde{S}-L|\tilde{S}_+\tilde{S}_-|\tilde{S}-L\rangle - L\right)\langle \tilde{S}|\tilde{S}^L_+\tilde{S}^L_-|\tilde{S}\rangle = \left(1 - \frac{L(L+1)}{2\tilde{S}}\right)\rho^0_{L,\tilde{S}\tilde{S}}, \tag{50}$$

(here we chose the proportionality coefficient in Eq. (47) to be 1; the particular choice of the coefficient does not affect the result for $\alpha_L$). We thus find

$$\alpha_L = \frac{L(L+1)}{2}. \tag{51}$$

Then, substituting Eq. (48) into Eq. (46) and using the fact that matrix $W_{n\to m}$ is symmetric we obtain

$$\Lambda_L = \frac{L(L+1)}{2\tilde{S}} \sum_{k=0}^{2\tilde{S}} k W_{\tilde{S}\to\tilde{S}-k} + O\left(\frac{1}{\tilde{S}^2}\right). \tag{52}$$

Here we implicitly assumed quick decay of coefficients $W_{\tilde{S}\to\tilde{S}-k}$ with $k$. Neglecting the corrections of order of $1/\tilde{S}^2$ we arrive to Eq. (32) of the main text.

## C   Numerical procedure

Here we provide the details of the numerical procedure used to solve master equation (13) and produce the plots presented in Fig. 3 and Fig. 4.

Generally, the problem of driven-dissipative dynamics of interacting particles in the magnetic field is tremendously complicated computation-wise. To simplify it, we make two assumptions justified within the scope of our work. First of all, in our numerical calculations we assume that the population of states with finite interaction energy or with components in higher Landau levels can be neglected. In that case, the density matrix of the system is composed only of quasihole states (10) with different particle numbers $N$ (which is equivalent to $\mathcal{P}\rho\mathcal{P} = \rho$). The number of quasihole many-body states is small relatively to the full size of the Hilbert space of the problem. Thus, the computational cost is greatly diminished.

To further reduce the complexity, we work with the master equation in the angular momentum representation [see Eq. (34) and Eq. (35)]. The rotational invariance evident in this

representation effectively reduces the number of non-zero components of the density matrix thus substantially cutting down the computation cost. To demonstrate how this works, we classify all basis states by $z$-projection of angular momentum, $S_z$. From Eqs. (34) and (35) it follows that the coherences between subspaces with different $S_z$ are not generated. Thus, if we assume that these coherences are zero initially then the dimensionality of the problem is reduced. Similar conclusion holds for coherences between states with different $N$. For example, if we throw away the coherences between states with different $S_z$ and $N$ for $N_\phi = 12$ ($N_{1/2} = 7$) the number of relevant components of the density matrix reduces from 372100 to 5530, i.e., by a factor of $\approx 70$.

Next, two crucial steps have to be made to solve the master equation. First of all, we need to construct an orthonormal basis of quasihole states with different particle number $N$ and angular momentum $S_z$. Second, we need to compute the matrix elements of annihilation operators $a_m$ between these states. We address both of these tasks using the formalism of Jack polynomials [69]. This formalism allows us to find the basis of quasihole states and expresses the corresponding basis vectors as superpositions of Fock states with different occupations of orbitals $\psi_m$ [defined in Eq. (3)]. Using the representation through Fock states it is straightforward to compute the matrix elements of the annihilation operators.

To introduce Jack polynomials we fix the number of flux quanta piercing the surface of the sphere, $N_\phi$, and the number of particles, $N$. At a given $N_\phi$ and $N$, the general form of the quasihole wave function [defined in Eq. (10)] can be equivalently rewritten as

$$\Psi_{\mathrm{qh}} = \left( \prod_i v_i^{N_\phi} \right) \tilde{P}(z_1, ..., z_N) \,, \tag{53}$$

where $z_i = u_i/v_i$ and $\tilde{P}(z_1, ..., z_N)$ is an arbitrary polynomial of degree no higher than $N_\phi$ in each coordinate that is (i) symmetric under permutations of $z_i$, (ii) vanishes for $z_i = z_j$, $i \neq j$. Jack polynomials (Jacks) $J_\lambda^{-1/2}$ provide a convenient set of linearly-independent polynomials $\tilde{P}$ that can thus be used to construct a basis of quasihole states (here $-1/2$ is an index specifying a particular type of Jacks). Different Jacks $J_\lambda^{-1/2}$ are labeled by the so-called $(2, 2, N)$-admissible partitions $\lambda$. A $(2, 2, N)$-admissible partition is an ordered set $\lambda = (q_1, q_2, ..., q_N)$ with $0 \leq q_i \leq N_\phi$ and $q_i > q_{i+1} + 1$. By calculating the total number of such partitions – and thus of Jacks – we can reproduce the number of quasihole states with fixed $N_\phi$ and $N$ [see Eq. (11)]. Notably, each Jack has a definite value of angular momentum,

$$S_z J_\lambda^{-1/2}(z_1, ..., z_N) = \sum_{i=1}^{N} q_i J_\lambda^{-1/2}(z_1, ..., z_N) \,, \tag{54}$$

where $S_z = \sum_{i=1}^{N} S_z^i$ with $S_z^i$ defined in Eq. (6). Therefore, since the prefactor in Eq. (53) possesses a well-defined angular momentum, $[S_z, \prod_i v_i^{N_\phi}] = -N N_\phi/2$, the classification of quasihole wave-functions by $S_z$ is equivalent to that of the corresponding Jacks (up to an offset of $-N N_\phi/2$). Wave-functions given by Eq. (53) that correspond to Jacks with different $S_z$ are orthogonal to each other. However, two independent Jacks with the same $N$ and $S_z$, in general, give rise to linearly dependent wave-functions. Thus, quasihole wave-functions with a given $N$ generated by Jacks with the same $S_z$ need to be orthogonalized.

The main property that makes Jack polynomials convenient for our purposes is that they have a known recursive expansion in terms of the monomials. Monomials are symmetric polynomials which are also labeled by partitions (though not necessary $(2, 2, N)$-admissible). They are defined as

$$\mathcal{M}_\lambda = z_1^{q_1} ... z_N^{q_N} + \text{permutations} \,, \tag{55}$$

where $0 \leq q_i \leq N_\phi$, $q_i \geq q_{i+1}$, and $\lambda$ denotes the partition $\lambda = (q_1, q_2, ..., q_N)$. Similarly to Jacks, monomials possess definite angular momentum $S_z = \sum_i q_i$. However, unlike Jacks, the

Table 1: List of $(2, 2, N)$-admissible partitions $\lambda$ for $N_\phi = 4$. These root partitions give rise to Jack polynomials with the corresponding value of $S_z$, see Eq. (54). Wavefunction Eq. (53) where $\tilde{P}$ is a given Jack has angular momentum $S_z - N N_\phi / 2$ (in particular, this implies that Laughlin state, $N = 3$, has vanishing angular momentum). The resulting wave-functions can be used to construct the basis of quasihole states with different $N$ for a given $N_\phi$. As a sanity check, we note that the total number of $(2, 2, N)$-partitions for a fixed $N$ is precisely equal to the number of quasihole states given by Eq. (11).

| $N$, particle number | $S_z$ of a Jack $J_\lambda^{-1/2}$ | $\lambda$, partition |
|---|---|---|
| 3 | 6 | $(4, 2, 0)$ |
| 2 | 6 | $(4, 2)$ |
| | 5 | $(4, 1)$ |
| | 4 | $(4, 0), (3, 1)$ |
| | 3 | $(3, 0)$ |
| | 2 | $(2, 0)$ |
| 1 | 4 | $(4)$ |
| | 3 | $(3)$ |
| | 2 | $(2)$ |
| | 1 | $(1)$ |
| | 0 | $(0)$ |

monomials are trivially related to Fock states with different occupancies of orbitals $\psi_m$ [see Eq. (3)]. Indeed, let us consider a combination

$$\Psi[\mathcal{M}_\lambda] = \left( \prod_i v_i^{N_\phi} \right) \mathcal{M}_\lambda(z_1, ..., z_N), \tag{56}$$

which is featured in the expansion of wave function (53) into the monomials. In terms of the Fock states we represent

$$|j_{N_\phi/2}, ..., j_{-N_\phi/2}\rangle = \frac{1}{\sqrt{N!}} \left( \prod_m \mathcal{C}_m^{j_m} \sqrt{j_m!} \right) \Psi[\mathcal{M}_\lambda], \tag{57}$$

where $\mathcal{C}_m$ is defined in Eq. (4), $j_m$ is the number of times $m + N_\phi/2$ is featured in the partition $\lambda$ (i.e., how many times an orbital with angular momentum $m$ is occupied), and the product runs over all $m = -N_\phi/2, ..., N_\phi/2$. Relation (57) allows us to rewrite the quasihole wave function in terms of Fock states and normalize it when the expansion of the corresponding Jack into the monomials is known. The expansion of wave functions into Fock states also allows us to perform easily the orthogonalization of degenerate subspaces of quasihole states with the same angular momentum and compute the matrix elements of the creation/annihilation operators between the quasihole states.

Now we describe how to expand a given Jack polynomial $J_\lambda^{-1/2}$ into a sum of monomials. The core operation to this end is the squeezing of partitions [69]. Squeezing changes the partition from $\mu = (q_1, ..., q_i, ..., q_j, ..., q_N)$ to $\mu' = (q_1, ..., q_i - t, ..., q_j + t, ..., q_N)$, where $t$ is an integer number satisfying $0 < t \leq |q_i - q_j|/2$. Note that by definition the partition is a non-growing set of numbers. Therefore, after the squeezing the partition might have to be reordered. Importantly, the squeezing does not change the angular momentum corresponding to the partition (since the latter is given by $\sum_i q_i$). The operation inverse to squeezing is the unsqueezing of the partition. It maps $\mu = (q_1, ..., q_i, ..., q_j, ..., q_N)$ to $\mu' = (q_1, ..., q_i + t, ..., q_j - t, ..., q_N)$ (with a proper reordering).

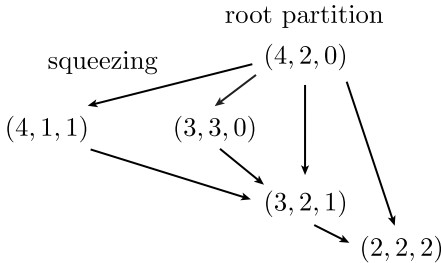

Figure 6: Directed graph formed by subsequent squeezing of a root partition $\lambda = (4, 2, 0)$. The resulting partitions determine which monomials are featured in Jack $J_{(4,2,0)}^{-1/2}$.

Next, there are two important mathematical statements that we will use, see, e.g., [69]. First of all, the only monomials $\mu$ that are featured in the expansion of $J_\lambda^{-1/2}$ are the ones that correspond to partitions obtained from the root partition $\lambda$ by a sequence of squeezing operations. While $\lambda$ has to be $(2, 2, N)$-admissible (i.e., satisfy $q_i > q_{i+1} + 1$), this is not the case for partitions $\mu$ [since squeezing operation can make the partition fall from the $(2, 2, N)$-admissible class]. This implies that the partitions involved in the expansion of a given Jack $J_\lambda^{-1/2}$ into the monomials form a tree-like structure (see Fig. 6 for an example). Specifically, the expansion of Jacks into the monomials can be written as [69]

$$J_\lambda^{-1/2}(z_1, ..., z_N) = \sum_{\mu \leq \lambda} c_{\lambda\mu} \mathcal{M}_\mu(z_1, ..., z_N), \tag{58}$$

where $\mu \leq \lambda$ indicates that $\mu$ can be obtained from $\lambda$ by applying multiple squeezing operations. By definition $c_{\lambda\lambda} = 1$.

The next important statement is that the coefficients can be calculated recursively [70,71]:

$$c_{\lambda\mu} = \frac{4}{l_\mu - l_\lambda} \sum_{\mu < \zeta \leq \lambda} \left[ (q_i + t) - (q_j - t) \right] c_{\lambda\zeta}. \tag{59}$$

Here, $\zeta = (\dots, q_i + t, \dots, q_j - t, \dots)$ denotes partitions that can be *directly* unsqueezed from $\mu = (q_1, \dots, q_N)$ and at the same time can be obtained by a sequence of squeezings from $\lambda$. Function $l_\mu$ can be found as

$$l_\mu = \sum_{i=1}^{N} q_i (q_i - 1 + 4(i - 1)), \tag{60}$$

(and similarly for $l_\lambda$). Eqs. (57), (58), (59), and (60) allow to find the expansion of wave functions into monomials explicitly.

Overall, our algorithm for solving master equation (13) can be summarized as follows:

1. Choose $N_\phi$ and the set of relevant particle numbers for which quasihole states are to be found. To simulate full dynamics we need to consider $N = 0, ..., N_{1/2}$ for even $N_\phi$ [where $N_{1/2} = N_\phi/2 + 1$] and $N = 0, ..., N_{1/2}^{\text{odd}}$ for odd $N_\phi$ [where $N_{1/2}^{\text{odd}} = (N_\phi + 1)/2$]. To study the dynamics of a single quasihole (see Section 4.1) we restrict attention to $N = N_{1/2}^{\text{odd}} - 1, N_{1/2}^{\text{odd}}$.

2. For each $N$ from the chosen set determine all possible $(2, 2, N)$-admissible partitions. Resulting partitions are root partitions for Jacks corresponding to different quasihole states. Notably, the root partitions fully determine the angular momentum of the state (see Table 1 for an example).

3. For all root partitions, expand wave functions (53) corresponding to all Jacks into the Fock states. To do that, use Eqs. (58), (59), and (60) to first expand into the monomials and Eq. (57) to convert from monomials to Fock states.

4. Resulting states with different $S_z$ or $N$ are by default orthogonal. States with the same $S_z$ and $N$ need to be orthogonalized since Jacks with same $S_z$ and $N$ generally give rise to non-orthogonal wave functions. When this is done a basis of quasihole states is formed.

5. Compute matrix elements of creation and annihilation operators $a_m$ and $a_m^\dagger$ corresponding to different orbitals $\psi_m$ between the quasihole states. Notably, there is a simple selection rule $\langle N, S_z | a_m | N' S_z' \rangle \propto \delta_{N+1,N'} \delta_{S_z+m,S_z'}$ (where $|N, M_z\rangle$ denotes any of the states with a given $N$ and $S_z$).

6. Solve the master equation (13) assuming that initially there are no coherences between states with different $S_z$ or $N$. The latter is to cut the computational cost. In principle, this assumption can be relaxed.

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
