# Peer review of "Stabilizing the Laughlin state of light: dynamics of hole fractionalization"

_SciPost Physics, doi:SciPost Phys. 13, 107 (2022)_

## Round 2 · Referee Report · Erich Mueller (Referee 1) · 2022-1-23

Strengths

1. Clearly identifies a problem
2. Produces a minimal model for exploring it
3. Thoroughly solves the model
4. The results have important consequences

Weaknesses

1. The primary results are negative
2. Introduction is overly pessimistic

Report

There is intense interest in creating exotic and strongly correlated states of light. One challenge is that optical systems are inevitably lossy. This has motivated a class of protocols where one actively pump the system, re-injecting photons into it, continually healing the damage done by the loss events. Here the authors explore an example, determining the limitations of using a local photon injection scheme to stabilize the bosonic Laughlin state against particle loss.

This Laughlin state supports “fractionalized” excitations, with unusual quantum statistics. It would be very exciting to have new platforms to explore this physics, and any progress towards this goal is valuable. Remarkably, the authors find that the very possibility of these non-local excitations poses an impediment to some stabilization schemes. In particular, they argue that photon loss and injection processes can lead to the creation of fractionalized excitations, and that these defects cannot be easily healed by local interventions. All stabilization strategies will need to contend with this challenge. In particular, local photon injection is at best an inefficient scheme for producing or stabilizing the Laughlin state.

The analysis used in the paper is well-conceived. They write down a minimal model for the dynamics, including both local photon loss, and local photon injection. They analyze the resulting dynamics, using clear and creative arguments. They also explore a number of ways of improving the stabilization efficiency.

I think this is an important step towards developing protocols for stabilizing exotic many-body phases of light. The principles developed here seem widely applicable -- they will be important for creating any state with fractionalized excitations. It is particularly appealing that the same features which make the state interesting are the ones which introduce impediments.

My sole negative comment is that the introduction of the paper has a somewhat more pessimistic tone than necessary. This is not the only approach to stabilization: Photon injection need not be local; Extra potentials can control the motion of quasiparticles; Boundaries can provide places for the quasiholes to be annihilated. By identifying hurdles, one can design future protocols which circumvent them.

In conclusion, I strongly endorse publishing this paper – though I am not sure that it fully fits any of the stated “acceptance criteria”. Rather than being a “groundbreaking discovery” or a breakthrough on a “stumbling block” – it is the identification of such a stumbling block, and an insightful observation that a community needs to change direction. It is, however, very important, and there is certainly potential for multi-pronged follow-up.

I find that it meets all of the “general acceptance criteria”. In particular I find it to be well written and self-contained.

Requested changes

1. The authors may find it useful to further emphasize positive aspects of the outlook.

  • validity: top
  • significance: high
  • originality: high
  • clarity: high
  • formatting: excellent
  • grammar: excellent

Author:  Pavel Kurilovich  on 2022-06-13  [id 2576]

(in reply to Report 1 by Erich Mueller on 2022-01-23)

We thank the referee for the positive assessment of our work, as well as for a thorough reading of the paper and thoughtful comments on our results. In particular we are thankful for suggesting us to give a more positive outlook to the manuscript.

We reformulated a paragraph in the introduction to make the tone less pessimistic (last paragraph on page 4 in the new version). We emphasized that our results present a hurdle which can potentially be overcome with further engineering efforts.

We also added a paragraph to the end of section 3 which emphasizes that the scaling of the quasihole number is only a power law despite the fractionalization. From this we conclude that the stabilization of the Laughlin state should in principle be feasible, in line with previous works on the dissipative stabilization.

---

## Round 2 · Referee Report · Anonymous (Referee 2) · 2022-2-16

Strengths

1. Timely theoretical investigation of one among the hottest challenges of contemporary quantum optics

2. Work of great interest for a broad spectrum of physicists from cond-mat and open quantum systems

3. Generally clear and exciting presentation

Weaknesses

1. Some specific points of the presentation needs clarification

2. Referencing to previous literature needs being improved

Report

The manuscript by Kurilovich reports a theoretical assessment of the feasibility of driven-dissipative stabilization schemes for a Laughlin states of light. In particular, the manuscript provides a thorough analysis of the role of hole fractionalization in slowing down the dynamics by trapping the many-body state of the system into dark (or quasi-dark) states that can not be further replenished by direct photon injection processes.
Overall, I liked very much reading this manuscript that combines high relevance and a remarkable mastering of technical tools with a very pleasant presentation of the results that clearly communicates the underlying physics. I may anticipate this work will have a major impact on the specific community of strongly correlated photon states (one of the holy grails of contemporary quantum optics) and has the potential to stimulate interesting developments also on neighboring fields of synthetic quantum matter, condensed matter physics and open quantum systems. The newly predicted mechanism for QH diffusion due to driving and dissipation is very intriguing and unexpected. With proper revisions, it will most likely make a wonderful SciPost Physics paper.

Before I can recommend publication on SciPost, I warmly invite the authors to take into due account the following important remarks:

1- I agree with the other reviewer that some parts of the text are too pessimistic and do not really match the actual conclusions of the work. I agree that fractionalization introduces additional difficulties compared to the recently observed Mott-insulator state, but none of the authors' conclusions appears to be an unsurmountable hurdle in the direction of a photon FQH state. The fact that both the higher QH density and the slower preparation time keep scaling polynomially in the parameters is somehow relieving (previous schemes, e.g., the one in Ref.37 had instead much worse scalings with N). Given that optical systems are typically very fast, I am not too much disturbed by the fact that relaxation may take long (in comparison with the very fast microscopic time-scales). I am afraid that other issues that are only briefly mentioned in this work, such as the chain of inequalities on the driving/dissipation parameters given on the 11th line of pag.23 may impose way more serious challenges to the experimentalists. The authors should amend the presentation and make it consistent with their results.

2- Referencing to previous work must be carried out in a proper and fair way.
2a- The idea that fractionalization may be a difficulty was already mentioned in [47], even though it was not fully developed there. On pag.3, the authors should explicitly recognize this when introducing fractionalization.
2b- Much of the discussion of Sec.3 was already present (from a slightly different perspective and, perhaps, in a not as clear way) in Ref.[49], where the key role of the quasi-hole degeneracy was fully taken into account. This scientific priority should be properly acknowledged. An explicit comparison of the final results of the two works should also be made, especially if the authors disagree with the conclusions of [49].

3- The discussion about the classification of the single-QH eigenstates in terms of angular momentum in Sec.4.1 is somehow obscure. My naive way of constructing a basis of such states would be to consider the classical parameters (u_0,v_0) as sort of quantum coordinates (with all caveats due to the over-completeness of the basis) and then build suitable linear superpositions of the many-body wavefunctions Psi[u0,v0]({ui,vi}) with a weight proportional to Y_{S,S_z}(u0,v0), that is PHI_{S,S_z}({ui,vi})=int du0 dv0 Y_{S,S_z}(u0,v0) Psi[u0,v0]({ui,vi}).
Is this picture correct? If so, it would be beneficial to include it in the revised manuscript. This would strongly help readers to follow the discussion.

4- I have trouble understanding the reasonings underlying the calculation of the relaxation eigenvalues \Lambda_L in the main text and in Appendix A. This is the crucial result of the manuscript and the authors should make a willing effort to clarify the presentation and make it understandable by generic readers.

5- I am curious to understand the functional form of the decay of W with k shown in fig.3(b). I guess it is exponential. To show this, could it be possible and useful to display (e.g. as an inset) the same plot in semilogy scale ?

6- On pag.18, second paragraph: the authors discuss the possibility of getting stuck into the N_{1/2}-1 manifold, but don't mention the possibility of getting stuck earlier in one of the N_{1/2}-j, j>1 manifolds. I agree that such manifolds do not host perfectly dark states, still some states may have an exponentially suppressed brightness which could be equally annoying. Some comments in this direction are needed: do such state change the scaling of the characteristic relaxation rate \propto \kappa? If possible, it would be also interesting to highlight signatures of such states in the time-dependent plot of fig.4(b).

7- The reaction-diffusion model in eq.(33) is very interesting. Since it has the potential to dramatically facilitate modeling of the dynamics, it is of crucial importance to assess its quantitative accuracy. To this purpose, the authors should display some comparison of its predictions with full numerical calculations, e.g. by adding the corresponding curve to fig.4(b) and/or by adding a newly designed figure.

8- The discussion in Sec.5.3 should be compared to the analysis in previous works, e.g. Ref.49. Are the conclusions in agreement?

9- A naive curiosity: in Sec.5.5 the authors say that "non-contractible loops around the torus" spoil the topological quantum coherence in the topologically degenerate manifold. Why don't they mention simple loops of one QH around another QH ? these latter are less dangerous to the coherence or occur less frequently? Some brief comment is needed.

10- In Sec.5.1, the authors discuss the effect of localized impurities on the relaxaton rate. How would their conclusion change if a smooth yet non-harmonic trapping potential was added to the system? Could this help to reinforce the relaxation rate wile avoiding issues from Anderson localization?

11- Minor typos: In the caption of Fig.3 "hoping rates" -> "hopping rates". Fig.4(b) shows the average photon number, but the caption incorrectly speaks about deviation from Laughlin.

Requested changes

1. Provide fair referencing to earlier literature

2. Clarify the presentation, in particular on points 3 and 4 above

3. Take into account all other points

  • validity: top
  • significance: top
  • originality: high
  • clarity: high
  • formatting: perfect
  • grammar: perfect

Author:  Pavel Kurilovich  on 2022-06-13  [id 2575]

(in reply to Report 2 on 2022-02-16)

We thank the referee for the positive assessment of our manuscript. We also thank the referee for a thorough reading of the paper, for detailed feedback, and for helpful suggestions.

1 - To give the text a more positive outlook, we rewrote the paragraph in the introduction about the implications of our results. The new paragraph (second paragraph on page 5) emphasizes that the fractionalization of quasiholes can potentially be overcome with additional engineering (e.g. by adding some sort of quasihole trapping potential). For the same reason, we added a paragraph about the experimental feasibility of the stabilized Laughlin state (second to last paragraph in Section 3 in the new version of the manuscript). There, we stress that the scaling of the number of missing particles in the steady state scales rather mildly with \kappa/\Gamma (although the scaling is worse than in a Mott insulator state).

We modified the text inspired by the referee's comment about the time scales of optical systems being “quick” (which implies that even though the relaxation dynamics is governed by the loss rate as opposed to the refilling rate, it can still be fast enough to run the experiments). To this end, we removed the emphasis on the slowness of the relaxation throughout the text.

Finally, we emphasized the importance of experimental challenges coming from the inequality mentioned by the referee. To this end we promoted the section of the conclusion devoted to the off-resonant injection of photons to be the first one in Section 5 (now it is Section 5.1 instead of 5.3).

2 - We added a note saying that the possible effect of hole fractionalization was mentioned in Ref. [47] in the introduction. We also added additional references and comparisons to Ref. [49]. In particular, after equation (17) we explicitly say that a similar equation was derived in Ref. [49]. Additionally, in the section of the conclusion about the off-resonant photon injection (Section 5.1 in the new version) we say that one of our estimates was also given in Ref. [49]. We also mentioned that the general conclusion about the experimental feasibility of the Laughlin state stabilization is in line with previous works Refs. [47-49] (second to last paragraph in Section 3).

3 - We thank the referee for pointing us to the incompleteness of our description of the basis states with a given angular momentum projection. We added explicit expression for the quasihole wavefunction with a given angular momentum projection (see Eq. (22) in the new version of the manuscript). With that explicit construction at hand we think that the equation proposed by the referee wouldn’t make the text clearer.

4 - Following the remark of the Referee, we introduced several changes to Section 4.1 and to the appendix to make the discussion more tractable.

First of all, we subdivided Section 4.1 into subsections 4.1.1 and 4.1.2. Subsection 4.1.1 is entirely devoted to the construction of the single-quasihole wavefunction. Subsection 4.1.2 discusses the relaxation rates that characterize the dynamics of a quasihole.

Second, we rewrote a substantial part of Section 4.1 to clarify the derivations. To this end, we explicitly introduced the concept of a relaxation eigenmode (see Eq. (25) and discussion around it). We also added several passages which motivate the steps in the calculation of the relaxation rates. In particular, we elaborated on our use of a classical rate equation (Eq. (29) in the new version), see paragraph after Eq. (28).

Third, we introduced a new section of the appendix, Appendix A in the new version. This section is devoted to deriving the relaxation eigenmodes from spin commutation relations.

Finally, we modified the appendix devoted to the derivation of relaxation rates for modes with different angular momenta (now Appendix B) to make it more comprehensible.

5 - In fact the rates W decay with k quicker than exponentially for each particular photon number. However, from Fig. 3 it is clear that the falloff becomes more gradual as the photon number is increased. This effect is especially pronounced in the tail of the falloff. Our main hypothesis is that the quicker-than-exponential decay is a finite size artifact which should converge to an exponential decay in the thermodynamic limit. The reason for this expectation is that the quasihole wave function on a plane is r^k exp(- r^2 / r_0^2) (by analogy with a single-particle Landau level problem). This wave function is centered at r \propto \sqrt(k) and as such the power in the exponent is linear in k. Therefore, if we compute the rates via a Fermi golden rule we should see the exponential decay with k. We added a footnote at page 17 which says that the decay of W is quicker than exponential. We refrain from adding the inset with a semi-log plot to Figure 3b since it already has an inset.

6 - We would like to point out to the referee that we have a brief comment about fractionalized configurations in manifolds with N_{1/2} - j particles, see page 19 paragraph 1 (“We note that…”). Overall, we believe that in a large system - such that it is possible to have several well-isolated quasiholes - the role of states with exponentially suppressed brightness starts to dominate the dynamics of the system. We think that in this case the relaxation rate is captured by our reaction-diffusion model, equation (37). This model predicts 1 / t_{relax} ~ \kappa^{3/2} / \Gamma^{1/2} while for a small system we get 1 / t_{relax} ~ \kappa. Therefore, the scaling of the relaxation rate becomes different, as suggested by the referee. We added two sentences to paragraph 1 on page 19 to emphasize the importance of approximate dark states with exponentially suppressed refilling. At the same time, we cannot capture this behavior by using the numerical solution of master equation (13). This is because we can only go up to particle numbers N_{1/2} ~ 10. In this case four quasiholes present at N = N_{1/2} - 2 cannot be separated far from each other and thus the exponential suppression of the refilling is not pronounced. In an attempt to find the states with suppressed refilling we looked at the relaxation rates for the refilling superoperator given by Eq. (15) truncating the Hilbert space to N_{1/2}, …, N_{1/2} - j particles. We find that for N_{1/2} = 9 and j = 1 the non-zero relaxation rates are all greater or equal to \approx 0.265 \Gamma. For N_{1/2} = 9 and j = 2 they are all greater or equal to \approx 0.197 \Gamma. This suggests that the subspace with N_{1/2} - 2 particles might host states with a suppressed relaxation although this suppression is only weak.

7 - In accord with our reply to point 6, we think that it is impossible to check how well Eq. (37) [in the new version] describes the behavior of master equation (13) using our computational capabilities. So at that point the applicability of Eq. (37) in the limit of a large system is only a conjecture. We added a paragraph about the comparison between Eq. (13) and Eq. (37) at the end of Section 4.2.

8 - Basic conclusions are in agreement, such as the rate at which high energy configurations are refilled. We added a reference to Ref. [49] after the estimate of this rate.

9 - Taking one quasihole around another will result in the overall phase factor common for all of the topologically degenerate states. Therefore, the processes mentioned by the referee are not important for the relaxation of the system.

10 - Disorder potential was considered merely as a simple example of a perturbation that induces the quasihole mobility. Other such perturbations are listed in the beginning of Section 5.2 (in the new version of the manuscript). We think that adding a smooth non-harmonic trapping - as suggested by the referee - generally reinforces the relaxation. For example, lattice potential is known to endow Landau levels with a finite velocity thus making the quasiholes mobile. This results in a finite relaxation rate even in the absence of loss (when there is no dissipative dynamics of the quasiholes).

11 - We thank the referee for pointing us to these typos. We corrected them in the new version of the manuscript.

---

## Round 3 · Referee Report · Anonymous (Referee 2) · 2022-6-28

Strengths

  1. Timely theoretical investigation of one among the hottest challenges of contemporary quantum optics

  2. Work of great interest for a broad spectrum of physicists from cond-mat and open quantum systems

  3. Generally clear and exciting presentation

Weaknesses

  1. None

Report

The authors have satisfactorily addressed all my remarks. I recommend the manuscript for publication.

Requested changes

  1. None

---

## Round 3 · Author Response

We thank the referees for positive assessment of our work and for the detailed feedback. We modified the text taking into the account comments, questions, and concerns of the referees.

---

## Round 3 · List of Changes

• Promoted the section of the conclusion devoted to the off-resonant injection of photons to be the first one in Section 5 (now it is Section 5.1 instead of 5.3).
  • Added a paragraph about the experimental feasibility of the stabilized Laughlin state (second to last paragraph in Section 3 in the new version of the manuscript).
  • Removed the emphasis on the relaxation dynamics being “slow” and instead emphasized that this dynamics is governed by the loss rate as opposed to the naively expected refilling rate.
  • Added a note saying that the possible effect of hole fractionalization was mentioned in [47] (end of the paragraph starting on page 3 and ending on page 4 in the new version).
  • Added a reference to Ref. [49] after equation (17).
  • Added a reference to Ref. [49] in Sec. 5.1.
  • Added explicit expression for the quasihole wavefunction with a given angular momentum projection (see Eq. (22) in the new version of the manuscript).
  • Subdivided Section 4.1 into subsections 4.1.1 and 4.1.2.
  • Rewrote a part of section 4.1 following the comments of the second Referee.
  • Added a new section of the appendix, Appendix A in the new version. This appendix provides the derivation of the relaxation eigenmodes.
  • Modified the appendix devoted to the derivation of relaxation rates for modes with different angular momenta (now appendix B) to make it more comprehensible.
  • Added a footnote about the dependence of the hopping rates on the angular momentum at page 17.
  • Added two sentences to paragraph 1 page 19 to emphasize the importance of approximate dark states with exponentially suppressed refilling.
  • We added a paragraph about the comparison between Eq. (13) and (37) at the end of Section 4.2.
  • We corrected the typos mentioned by the second referee.
  • Following the comments of both referees, we reformulated one paragraph in the introduction to make the tone less pessimistic (second to last paragraph in Section 1). We emphasized that our results present a hurdle which can potentially be overcome with further engineering efforts.
  • We added a paragraph to the end of Section 3 which emphasizes that the scaling of the quasihole number is only a power law despite the fractionalization. From this we conclude that the stabilization of the Laughlin state should in principle be feasible, in line with previous works on the dissipative stabilization.

---

## Editorial Decision

published